# ▤ LADDER: LANGUAGE DRIVEN SLICE DISCOVERY AND ERROR RECTIFICATION

## ABSTRACT

Error slice discovery is crucial to diagnose and mitigate model errors. Current clustering or discrete attribute-based slice discovery methods face key limitations: 1) clustering results in incoherent slices, while assigning discrete attributes to slices leads to incomplete coverage of error patterns due to missing or insufficient attributes; 2) these methods lack complex reasoning, preventing them from fully explaining model biases; 3) they fail to integrate *domain knowledge*, limiting their usage in specialized fields *e.g.,* radiology. We propose LADDER (L̲anguage-D̲riven D̲iscovery and E̲rror R̲ectification), to address the limitations by: (1) leveraging the flexibility of natural language to address incompleteness, (2) employing LLM's latent *domain knowledge* and advanced reasoning to analyze sentences and derive testable hypotheses directly, identifying biased attributes, and form coherent error slices without clustering. Existing mitigation methods typically address only the worst-performing group, often amplifying errors in other subgroups. In contrast, LADDER generates pseudo attributes from the discovered hypotheses to mitigate errors across all biases without explicit attribute annotations or prior knowledge of bias. Rigorous evaluations on 6 datasets spanning natural and medical images – comparing 200+ classifiers with diverse architectures, pretraining strategies, and LLMs – show that LADDER consistently outperforms existing baselines in discovering and mitigating biases. The code is available[1].

## 1 INTRODUCTION

Discovering error slices in models is essential for mitigating their limitations. Existing slice discovery methods typically cluster misclassified samples by similar attributes or directly analyze model biases using a set of predefined attributes. Although these methods are intuitive and support straightforward mitigation like rebalancing, they face several key issues: 1) unsupervised clustering approaches often produce incoherent attribute groupings, while direct attribute-based methods suffer from incompleteness due to missing or insufficient attribute annotation; 2) they lack reasoning capability about the deeper complexities of model errors; and 3) they do not incorporate essential *domain knowledge* critical in specialized fields, *e.g.,* radiology. In contrast, our approach, LADDER, is the first to use LLMs for slice discovery beyond simple *keyword-based attribute searches* and addresses these issues by (1) using the flexibility of natural language rather than merely relying on the presence/absence of attributes; (2) leveraging the reasoning capabilities and *domain knowledge* of LLMs to identify coherent error slices from language without relying on unsupervised clustering.

Language-aware slice discovery methods *e.g.,* DrML (Zhang et al., 2023), leverage language through the text encoder to mitigate biases in the CLIP vision encoder by closing the modality gap with cross-modal transferability. However, this reliance on cross-modal transfer hinders their applicability to non-multimodal models. Also, DrML relies on user-defined prompts, introducing subjectivity and potential human bias into

---

[1] https://github.com/AI-annonymous/ICLR-submission

the error rectification process. Similarly, Facts (Yenamandra et al., 2023) amplifies the spuriousness in the initial training stage by setting large weight decay, deviating from standard supervised learning practices. Methods like Domino (Eyuboglu et al., 2022) and Facts discover slices by clustering samples with similar attributes within the vision-language representation (VLR) space. However, the slices often exhibit semantic inconsistencies – attributes within slices lack coherence, leading to unreliable interpretations of model errors. PRIME (Rezaei et al., 2023) relies on expensive tagging models (Zhang et al., 2024), limited to detecting the presence or absence of attributes. HiBug (Chen et al., 2024) prompts LLMs for potentially biased attributes (via *keywords*) based solely on general user prompts without incorporating any textual context from the dataset itself. These tags or attributes can be incomplete. Also, these methods lack reasoning capabilities and *domain knowledge* needed for complex error patterns, failing to capture relevant biases for specialized tasks.

Existing bias mitigation methods *e.g.,* GroupDRO (Sagawa et al., 2020), JTT (Liu et al., 2021), DFR (Kirichenko et al., 2022) rely on expensive and often incomplete attribute annotations in the training or validation sets. While these methods improve the performance of the worst-performing group, they can inadvertently amplify errors in other groups, highlighted by Li et al. (2023b). Although Li et al. (2023b) addresses errors across multiple biases, it assumes prior knowledge about the number and types of biases to design specific data augmentations. This reveals a critical gap: the need for an automated method for discovering slices from data and mitigating multiple biases w/o prior knowledge or annotations.

**Contributions.** This paper introduces LADDER to address the gaps in slice discovery and bias mitigation. It detects slices on any off-the-shelf supervised classifier, overcoming the specific training requirements of Facts and DrML. Unlike Domino and Facts, which project images directly into VLR space, LADDER projects the classifier's representations to VLR space to preserve semantic coherence. Motivated by language-driven localization (Zhong et al., 2022; Yu et al., 2022), LADDER uses image captions or radiology reports to retrieve sentences indicative of model errors, utilizing the flexibility of natural language to capture deeper insights beyond the simple presence or absence of attributes, unlike tagging models. It then leverages LLM's reasoning capability to generate testable hypotheses that identify biases leading to classification errors. To illustrate, we train a classier on a synthetic dataset (Appendix A.11) where Class 0 images consistently have a yellow box to the left of a red box (Fig 1), introducing a spurious correlation based on relative positioning. The classifier exhibits poor performance on test data without the bias. The

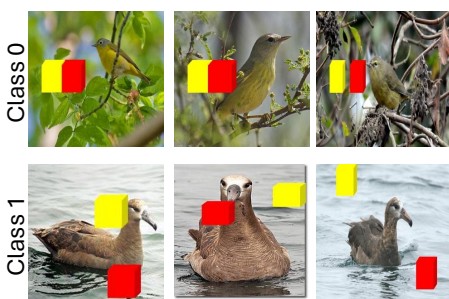

Figure 1: Synthetic dataset featuring Class 0 images consistently with a yellow box to the left of a red box. In Class 1 images, the boxes are randomly placed.

LLM analyzes the sentences to generate hypotheses (Fig 7), correctly identifying the classifier's bias towards box positioning. Thus LADDER can uncover complex biases beyond tagging models' capabilities. Note, LADDER invokes LLM only once with the text tokens only. For mitigation, LADDER generates pseudo-labels for the biased attributes corresponding to each hypothesis and fine-tunes the linear head of the classifier either by reweighting or rebalancing. We use an ensemble approach to derive predictions from the debiased model for each hypothesis, thereby mitigating multiple biases without explicit attribute annotations or prior knowledge of their number and type. Rigorous evaluations on six datasets across various architectures and pretraining strategies illustrate that LADDER outperforms baselines in discovering and mitigating biases.

## 2 LADDER

**Notation:** Assume the classifier $f = g \circ \Phi$ is trained using ERM to predict the labels $\mathcal{Y}$ from the images $\mathcal{X}$, where $\Phi$ and $g$ are the representation and classification head, respectively. $\{\Psi^I, \Psi^T\}$ denote the image

and text encoders of the joint VLR space (*e.g.,* CLIP). For a set of images $\mathcal{X}_Y$ of a class $Y \in \mathcal{Y}$, LADDER finds error slices where $f$ underperforms and fixes it. It employs a text corpus $t_{val}$ from radiology reports or image captions. Note that we do not need paired image captions. Throughout the paper, $\langle \cdot, \cdot \rangle$ denotes the dot product to estimate the similarity between two representations. Fig. 2 shows the schematic of LADDER. We do not rely on sample-specific annotations, human-generated prompts, or prior knowledge of bias types or their numbers. LADDER utilizes the validation dataset (can be a small subset of training data, not used during training) to discover and mitigate errors.

**Error slice:** An error slice for a class $Y$ includes subsets $\mathcal{X}_Y$ where the model performs significantly worse than its overall performance on the entire class $Y$, formally defined as: $\mathbb{S}_Y = \{\mathcal{S}_{Y,\neg\texttt{attr}} \subseteq \mathcal{X}_Y | e(\mathcal{S}_{Y,\neg\texttt{attr}}) \gg e(\mathcal{X}_Y), \exists \texttt{attr}\}$, where $e(\cdot)$ is the error rate on the specific data subset and $\mathcal{S}_{Y,\neg\texttt{attr}}$ denotes the subset of $\mathcal{X}_Y$ without the attribute $\texttt{attr}$. Alternatively, $f$ is biased on the attribute $\texttt{attr}$, resulting in better performance on the subpopulation with $\texttt{attr}$ *e.g.,* error rate in pneumothorax patients without chest tubes is higher than that of overall pneumothorax patients (Docquier & Rapoport, 2012).

## 2.1 RETRIEVING SENTENCES HIGHLIGHTING MODEL'S ERROR

First, for a particular class, LADDER retrieves the sentences that describe the visual attributes contributing to correct classifications but missing in misclassified ones, leading to model errors. Following Moayeri et al. (2023a), it learns a projection function $\pi : \Phi \rightarrow \Psi^I$ (Appendix A.2 for details) to align the representation of the classifier, $\Phi$, with the image representation, $\Psi^I$ of the VLR space. Then, for a class label $Y$, we estimate the difference in mean of the projected representations of the correct and misclassified samples as $\Delta^I = \mathbb{E}_{X,Y|f(X)=Y}[\pi(\Phi(X)] - \mathbb{E}_{X,Y|f(X)\neq Y}[\pi(\Phi(X))]$. Assuming the mean representations preserve semantics, this difference captures key attributes contributing to correct classifications but are poorly captured or misrepresented in misclassified ones. Denoting the text embedding of $t_{val}$ as $\Psi^T(t_{val})$, we retrieve the $\texttt{topK}$ sentences as: $\texttt{topK} = \mathscr{R}(\langle \Delta^I, \Psi^T(t_{val}) \rangle, t_{val})$, where $\mathscr{R}$ is a retrieval function retrieving $\texttt{topK}$ sentences from the text corpus having the highest similarity score with the mean difference of the projected image representations. Next, the LLM analyzes the sentences and constructs hypotheses to find error slices.

## 2.2 HYPOTHESIS GENERATION VIA LLM AND DISCOVERING ERROR SLICES

**Hypothesis generation.** To retrieve the set of hypotheses, LADDER invokes an LLM with the $\texttt{topK}$ sentences. Formally, $\{\mathcal{H}, \mathcal{T}\} = \texttt{LLM}(\texttt{topK})$, where $\mathcal{H}$ is a set of hypotheses with attributes on which $f$ may be biased and $\mathcal{T}$ is a set of sentences to be used to test each hypothesis. $f$ underperforms on the subpopulation without the attributes in $\mathcal{H}$. Each hypothesis $H \in \mathcal{H}$ is paired with $\mathcal{T}_H \in \mathcal{T}$, a set of sentences that provide diverse contextual descriptions of the hypothesis-specific attribute as it appears in various images. Representations of images with the attribute specified in $H$, are highly similar to the mean text embedding of $\mathcal{T}_H$. Refer to Appendix A.6 for the prompt utilized by LLM to generate the hypothesis.

**Discovering error slices.** For each hypothesis $H \in \mathcal{H}$, we first compute the mean embedding of the set of sentences $\mathcal{T}_H$ as $\Psi^T(\mathcal{T}_H) = \frac{1}{|\mathcal{T}_H|} \sum_{t \in \mathcal{T}_H} \Psi^T(t)$. Now for an image $X \in \mathcal{X}_Y$, we obtain the projected representation $\pi(\Phi(X))$ in VLR space and compute the following similarity score,

$$s_H(X) = \langle \pi(\Phi(X)), \Psi^T(\mathcal{T}_H) \rangle \tag{1}$$

Finally, for a class label $Y$, we retrieve images with similarity scores below a threshold $\tau$ as $\mathcal{S}_{Y,\neg H} = \{X \in \mathcal{X}_Y | s_H(X) < \tau\}$. The hypothesis $H$ fails in these images as they lack the attribute specified in the $H$. The subset $\mathcal{S}_{Y,\neg H}$ may be a potential error slice, if the error $e(\mathcal{S}_{Y,\neg H})$ is greater than $\mathcal{X}_Y$. Formally, $\hat{\mathbb{S}}_Y$, the predicted slice for a class $Y$ is: $\hat{\mathbb{S}}_Y = \{\mathcal{S}_{Y,\neg H} \subseteq \mathcal{X}_Y | e(\mathcal{S}_{Y,\neg H}) \gg e(\mathcal{X}_Y), \exists H \in \mathcal{H}\}$

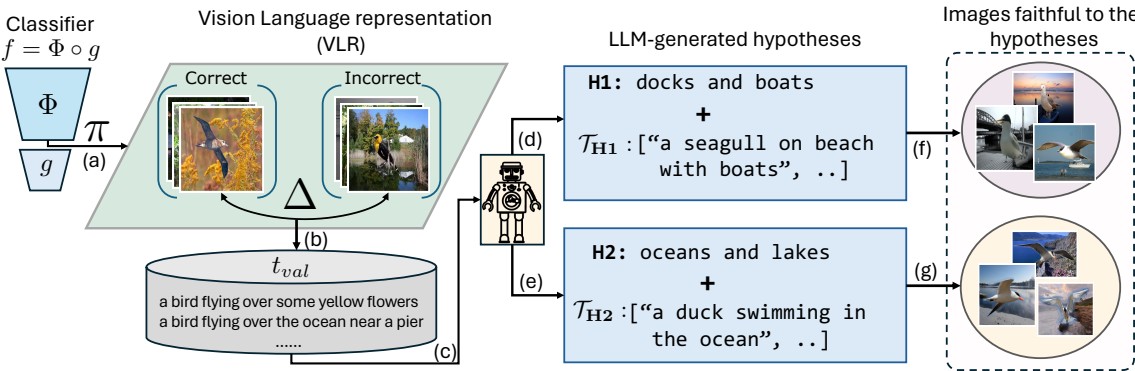

Figure 2: Schematic of slice discovery of LADDER. **(a)**: Projection ($\pi$) of model representation ($\Phi$) to VLR space. **(b)**: Retrieval of $\texttt{topK}$ sentences based on the difference in image embeddings ($\Delta$) within VLR space. **(c)**: LLM is invoked with $\texttt{topK}$ sentences. **(d-e)**: LLM generated hypotheses and sentences ($\{\mathcal{H}, \mathcal{T}\}$) to test the hypotheses. **(f-g)**: Finding the clusters faithful to the hypotheses.

## 2.3 MITIGATING ERRORS WITHOUT ANNOTATION

For the attributes linked to a hypothesis, LADDER treats $s_H$ as a logit and converts it to a probability. If the probability exceeds a threshold (0.5 in all experiments), LADDER assigns a pseudo-label 1 to the attribute and 0 otherwise. Thus, it generates pseudo-labels for all relevant attributes, enabling error mitigation without annotations. To do so, LADDER adopts an ensemble-based strategy, either reweighting or rebalancing. **Strategy 1: reweighting.** We assume the data-generating process for each hypothesis, where an attribute influences both the label $Y$ and the image $X$, and $Y$ subsequently influencing $X$. Inspired by IPTW (Austin, 2011) in causal inference, we apply a weight of $\frac{1}{P(Y|\text{attribute})}$ to each training sample and finetune the last layer $g$ of the classifier to upweight the minority samples. We calculate the weight using the pseudo attributes. We repeat this process for each hypothesis, resulting in a debiased model per hypothesis. During inference, we compute the similarity score $s_H(X)$ for all hypotheses and select the classifier head $g_{H^*}$ associated with the hypothesis that has the highest similarity score for the test image $X$, where $H^* = \arg\max_{H \in \mathcal{H}} s_H(X)$. We refer to this as LADDER$_{reweight}$. **Strategy 2: rebalancing.** Following DFR, we create a balanced dataset from a held-out validation set, for each pseudo-labeled attribute per hypothesis. We then fine-tune the classification head $g$ using this balanced dataset, producing a debiased model per hypothesis. During inference, we again compute the similarity score $s_H$ for all hypotheses and select the classifier head $g_{H^*}$ associated with the hypothesis having maximum similarity: $H^* = \arg\max_{H \in \mathcal{H}} s_H(X)$. We term this as LADDER$_{bal}$. Empirically, in our experiments, LADDER$_{bal}$ outperforms LADDER$_{reweight}$.

## 3 EXPERIMENTS

We perform experiments to answer the research questions: **RQ1.** How does LADDER perform in discovering error slices compared to other methods? **RQ2.** How does LADDER leverage latent medical knowledge and perform attribute-unconstrained identification with LLMs to enhance slice discovery? **RQ3.** How do the attributes in the LADDER-discovered hypotheses vary with different architectures and pre-training methods? **RQ4.** How does LADDER mitigate biases across benchmark datasets under different architectures and pre-training methods? **Additionally**, we further explore 1) the boost in zero-shot accuracy using attributes discovered by LADDER (Appendix A.13.8), 2) the CLIP score (Kim et al., 2024) for the attributes in the hypotheses (Appendix A.13.9), and 3) ablations on LADDER's performance with different captioning methods (Appendix A.13.13), the diversity of slices discovered with different LLMs (Appendix A.13.14), and bias

mitigation using different LLMs (Appendix A.13.15). **Note, across bias mitigation results, LADDER refers to LADDER$_{bal}$ unless specified**. Refer to Tab. 1 for an overview of the datasets used to evaluate LADDER.

Table 1: Evaluation datasets and tasks with the spurious correlations. Refer to the Appendix A.7 for details.

| Classification Task | Dataset | Modality | Spurious Correlation |
|---|---|---|---|
| Landbird vs. Waterbird (Wah et al., 2011) | Waterbirds | Natural image | Background (land or water) |
| Hair Color (blond vs. non-blond) (Liu et al., 2018) | CelebA | Natural image | Gender (94% blond are female) |
| Cat vs. Dog (Liang et al., 2022) | MetaShift | Natural image | Background (dogs outdoors, cats indoors) |
| Pneumothorax (Wang et al., 2017; Docquier & Rapoport, 2012) | NIH | Chest-X-Rays (CXRs) | Presence of chest tube |
| Breast Cancer (Wen et al., 2024) | RSNA-Mammo | 2D mammograms | Calcifications |
| Abnormality (Nguyen et al., 2023) | VinDr-Mammo | 2D mammograms (CXRs) | Calcifications |

Table 2: Experimental setup of evaluating LADDER. RN and EN mean ResNet50 and EfficientNet, respectively

| Modality | Architecture of classifier ($f$) | Image Size | VLR Space($\{\Psi^I, \Psi^T\}$) | # `topk` Sentences (Sec. 2.1) |
|---|---|---|---|---|
| **Natural Images** | RN (He et al., 2016),ViT (Dosovitskiy et al., 2020) | $224 \times 224$ | CLIP (ViT-B/32) (Radford et al., 2021) | 200 |
| **CXRs** | RN, ViT | $224 \times 224$ | CXR-CLIP (ViT-B/32) (You et al., 2023) | 100 |
| **Mammograms** | EN-B5 (Tan & Le, 2019) | $1520 \times 912$ | Mammo-CLIP (EN-B5) (Ghosh et al., 2024) | 100 |

**Experimental details.** Refer to Tab. 2 for the classifiers ($f$) LADDER aims to probe. For natural images and CXRs, we initialize $f$ using the pertaining methods such as supervised (Sup) (Kornblith et al., 2019), SimCLR (Chen et al., 2020), Barlow Twins (Zbontar et al., 2021), DINO (Caron et al., 2021), and CLIP-based (Radford et al., 2021) on datasets including ImageNet-1K (IN1k) (Deng et al., 2009), ImageNet-21K (IN21k) (Ridnik et al., 2021), SWAG (Singh et al., 2022), LAION-2B (Schuhmann et al., 2022), and OpenAI-CLIP (OAI) (Radford et al., 2021). So, **RN Sup IN1k** denotes a **supervised-ImageNet-1K pretrained ResNet50** classifier. For mammograms, $f$ is initialized with supervised IN1k weights. For the text corpus ($t_{val}$), we use BLIP-captioner(Li et al., 2022) for natural images and radiology reports from MIMIC-CXR (Johnson et al., 2019) for NIH. For mammograms, we use the radiology text from the language-driven weak-supervision task in Mammo-FActOR (Ghosh et al., 2024) (Appendix A.10.3). We use GPT-4o (Wu et al., 2024) as LLM to generate hypotheses. Error slices are defined as subsets where the error rate exceeds the overall class error by at least 10%. Further experimental and ablation details are in Appendix A.10. **Baselines.** For slice discovery, we compare LADDER with Domino and Facts(Appendix A.8 for details). For bias mitigation, we compare LADDER with 15 baselines (Appendix A.9 for details), including ERM (Vapnik, 1999), GroupDRO (Sagawa et al., 2020), CVaRDRO (Duchi & Namkoong, 2021), LfF (Nam et al., 2020), JTT (Liu et al., 2021), LISA, (Yao et al., 2022), DFR (Guo et al., 2019), Mixup (Zhang et al., 2018), IRM (Arjovsky et al., 2020), MMD (Li et al., 2018), Focal (Lin et al., 2017), CBLoss (Cui et al., 2019), LDAM (Cao et al., 2019), CRT (Kang et al., 2020), ReWeightCRT (Kang et al., 2020). **Evaluation metrics.** For slice discovery, we use `Precision@10` (Appendix A.3) (Eyuboglu et al., 2022) to evaluate the slice discovery methods and the CLIP score (Kim et al., 2024) to quantify the effect of biased attributes. Also, we propose `AccGap` (Appendix A.4) to compare the performance of slice discovery algorithms by evaluating a discovered slice (e.g., waterbirds without lake) against a closely related ground truth slice (e.g., waterbirds not on water). For error mitigation, we report Worst Group Accuracy (WGA) and mean accuracy for natural images, with WGA representing the accuracy of the worst-performing subgroup. For medical images, we report mean AUROC and WGA, where WGA refers to model performance on pneumothorax patients without chest tubes (NIH) and cancer or abnormal patients without calcifications (RSNA-Mammo, VinDr-Mammo).

# 4 RESULTS

**Comparison of LADDER with slice discovery baselines (RQ1).** Tab. 3 compares the `Precision@10` of different slice discovery methods for CNN models (EN-B5 for mammograms & RN Sup IN1k for others).

For medical images, LADDER outperforms the baselines($\sim$**50%**$\uparrow$). Next, we evaluate the quality of slices each method produces to determine their effectiveness in facilitating bias mitigation. More coherent slices result in more effective mitigation outcomes. So, we discover slices with Domino, Facts, and LADDER, apply our bias mitigation strategy (Sec. 2.3) by constructing balanced datasets based on the discovered biases, and compute the WGA. Fig. 3 reports the WGA and shows that LADDER outperforms the other slice discovery baselines across all experimental settings for Waterbirds, CelebA, and NIH. Refer to Appendix A.13.12 for mammograms. Facts and Domino cluster the images by projecting them directly into VLR space, often leading to incoherent slices. In contrast, LADDER first projects the model's representation, $\Phi^I$, into the VLR space, preserving the nuanced semantics of the classifier features. Instead of relying solely on unsupervised clustering, it leverages the reasoning capabilities of LLMs and signals from the captions/radiology reports to identify the coherent-biased attributes within the discovered slices. Next, we assign pseudo-labels to the attributes using similarity scores (Eq. 1). The coherent slices produced by LADDER ensure that the pseudo-labeling process is more accurate than the baselines, leading to superior bias mitigation performance.

Table 4: Benchmarking error mitigation methods over 3 seeds for CNN models (EN-B5 for mammograms and RN Sup 1k for the rest). We bold-face and underline the best and second-best results, respectively.

| Method | Waterbirds | | CelebA | | NIH | | RSNA | | VinDr | |
|---|---|---|---|---|---|---|---|---|---|---|
| | Mean(%) | WGA(%) | Mean(%) | WGA(%) | Mean(%) | WGA(%) | Mean(%) | WGA(%) | Mean(%) | WGA(%) |
| Vanilla (ERM) | $88.2_{\pm0.7}$ | $69.1_{\pm1.2}$ | $94.1_{\pm0.2}$ | $62.2_{\pm1.5}$ | $\mathbf{86.8}_{\pm0.0}$ | $60.3_{\pm0.0}$ | $\mathbf{85.3}_{\pm0.0}$ | $69.8_{\pm0.0}$ | $\mathbf{86.9}_{\pm0.0}$ | $45.6_{\pm0.0}$ |
| Mixup | $88.5_{\pm0.5}$ | $77.3_{\pm0.5}$ | $\underline{94.5}_{\pm0.1}$ | $57.8_{\pm0.8}$ | $85.1_{\pm0.0}$ | $67.6_{\pm0.8}$ | $84.5_{\pm0.0}$ | $64.8_{\pm0.0}$ | $83.2_{\pm0.0}$ | $65.3_{\pm0.0}$ |
| IRM | $88.1_{\pm0.2}$ | $74.3_{\pm0.1}$ | $\underline{94.5}_{\pm0.5}$ | $63.3_{\pm2.5}$ | $83.2_{\pm0.0}$ | $63.4_{\pm0.0}$ | $83.3_{\pm0.0}$ | $68.4_{\pm0.0}$ | $83.5_{\pm0.0}$ | $65.2_{\pm0.0}$ |
| MMD | $92.5_{\pm0.1}$ | $83.5_{\pm1.1}$ | $92.5_{\pm0.6}$ | $22.7_{\pm2.5}$ | $84.6_{\pm0.0}$ | $65.4_{\pm0.0}$ | $84.2_{\pm0.0}$ | $69.1_{\pm0.0}$ | $81.2_{\pm0.0}$ | $64.8_{\pm0.0}$ |
| Focal | $89.3_{\pm0.2}$ | $71.6_{\pm0.8}$ | $\mathbf{94.9}_{\pm0.3}$ | $59.3_{\pm2.0}$ | $85.5_{\pm0.0}$ | $68.9_{\pm0.7}$ | $83.6_{\pm0.0}$ | $65.5_{\pm0.0}$ | $82.6_{\pm0.0}$ | $63.7_{\pm0.0}$ |
| CBLoss | $91.3_{\pm0.7}$ | $86.1_{\pm0.3}$ | $91.2_{\pm0.7}$ | $89.3_{\pm0.5}$ | $85.5_{\pm0.0}$ | $63.4_{\pm0.0}$ | $83.2_{\pm0.0}$ | $65.1_{\pm0.0}$ | $81.7_{\pm0.0}$ | $62.5_{\pm0.0}$ |
| LDAM | $91.3_{\pm0.7}$ | $86.1_{\pm0.3}$ | $\underline{94.5}_{\pm0.2}$ | $58.3_{\pm2.5}$ | $84.3_{\pm0.0}$ | $69.4_{\pm0.2}$ | $81.6_{\pm0.0}$ | $63.5_{\pm0.0}$ | $81.2_{\pm0.0}$ | $62.2_{\pm0.0}$ |
| CRT | $90.5_{\pm0.2}$ | $79.7_{\pm0.3}$ | $92.5_{\pm0.1}$ | $87.3_{\pm0.3}$ | $82.7_{\pm0.0}$ | $68.5_{\pm0.0}$ | $82.7_{\pm0.0}$ | $68.8_{\pm0.0}$ | $82.9_{\pm0.0}$ | $63.3_{\pm0.0}$ |
| ReWeightCRT | $91.3_{\pm0.1}$ | $78.4_{\pm0.1}$ | $92.5_{\pm0.2}$ | $87.2_{\pm0.5}$ | $83.0_{\pm0.0}$ | $69.5_{\pm0.0}$ | $82.4_{\pm0.0}$ | $68.3_{\pm0.0}$ | $82.9_{\pm0.0}$ | $63.3_{\pm0.0}$ |
| JTT | $88.8_{\pm0.7}$ | $84.5_{\pm0.3}$ | $90.6_{\pm2.2}$ | $87.2_{\pm7.5}$ | $85.1_{\pm0.0}$ | $70.4_{\pm0.0}$ | $84.6_{\pm0.0}$ | $68.5_{\pm0.0}$ | $83.7_{\pm0.0}$ | $66.1_{\pm0.0}$ |
| GroupDRO | $88.8_{\pm1.7}$ | $87.1_{\pm1.3}$ | $91.4_{\pm0.6}$ | $\underline{88.1}_{\pm0.7}$ | $85.2_{\pm0.0}$ | $71.1_{\pm0.0}$ | $85.1_{\pm0.0}$ | $72.3_{\pm0.0}$ | $82.7_{\pm0.0}$ | $67.1_{\pm0.0}$ |
| CVaRDRO | $89.8_{\pm0.4}$ | $85.4_{\pm2.3}$ | $\underline{94.5}_{\pm1.5}$ | $83.1_{\pm1.5}$ | $85.7_{\pm0.1}$ | $71.3_{\pm0.0}$ | $85.4_{\pm0.0}$ | $71.7_{\pm0.0}$ | $82.7_{\pm0.0}$ | $67.1_{\pm0.0}$ |
| LfF | $87.0_{\pm0.3}$ | $75.2_{\pm0.7}$ | $81.1_{\pm5.6}$ | $63.0_{\pm4.4}$ | $75.9_{\pm0.0}$ | $61.6_{\pm0.0}$ | $79.8_{\pm0.0}$ | $66.4_{\pm0.0}$ | $82.4_{\pm0.0}$ | $64.5_{\pm0.0}$ |
| LISA | $\mathbf{92.8}_{\pm0.3}$ | $88.7_{\pm0.6}$ | $92.6_{\pm0.1}$ | $86.2_{\pm1.1}$ | $85.2_{\pm0.0}$ | $66.6_{\pm0.0}$ | $85.1_{\pm0.0}$ | $64.4_{\pm0.0}$ | $82.8_{\pm0.0}$ | $63.1_{\pm0.0}$ |
| DFR$_{val}$ | $\underline{92.3}_{\pm0.2}$ | $88.2_{\pm0.3}$ | $89.3_{\pm0.2}$ | $87.1_{\pm1.1}$ | $86.1_{\pm0.0}$ | $70.5_{\pm0.0}$ | $85.1_{\pm0.0}$ | $71.2_{\pm0.0}$ | $83.8_{\pm0.0}$ | $68.1_{\pm0.0}$ |
| LADDER$_{reweight}$ (ours) | $91.6_{\pm0.6}$ | $\underline{92.7}_{\pm0.6}$ | $89.8_{\pm0.9}$ | $88.4_{\pm0.4}$ | $\underline{86.5}_{\pm0.0}$ | $74.3_{\pm0.0}$ | $\underline{85.2}_{\pm0.0}$ | $\underline{74.7}_{\pm0.0}$ | $84.7_{\pm0.0}$ | $80.8_{\pm0.0}$ |
| LADDER$_{bal}$ (ours) | $92.1_{\pm0.8}$ | $\mathbf{93.7}_{\pm0.8}$ | $89.7_{\pm1.2}$ | $\mathbf{90.2}_{\pm0.4}$ | $86.3_{\pm0.0}$ | $\mathbf{76.2}_{\pm0.0}$ | $84.8_{\pm0.0}$ | $\mathbf{76.4}_{\pm0.0}$ | $\underline{86.2}_{\pm0.0}$ | $\mathbf{82.5}_{\pm0.0}$ |

**Leveraging Latent Medical Knowledge and Attribute-Unconstrained Identification with LLMs for Bias Detection (RQ2).** Fig.5 shows slice discovery by LADDER for classifying pneumothorax and waterbirds. In NIH (Fig.5a), LADDER detects subtle, domain-specific biases (*e.g.,* `chest tubes`, `size of pneumothorax`, `loculated nature` etc.) by retrieving sentences from correctly classified samples (Sec. 2.1). For RSNA-Mammo (Fig. 23), LADDER detects even the subtypes of `calcifications`, offering a more granular characterization of biases. This level of precision captures important medical insights that ML practitioners may overlook without radiology expertise. Unlike basic keyword extraction

Table 3: Precision@10 for CNN models ($f$).

| Dataset | Domino | FACTS | Ours |
|---|---|---|---|
| Waterbirds (Waterbird-Land) | 0.8 | 0.9 | **1.0** |
| Waterbirds (Landbird-Water) | **1.0** | **1.0** | **1.0** |
| CelebA (Blonde-Male) | 0.9 | 0.9 | 0.9 |
| MetaShift (Cat-Outdoor) | 0.5 | **0.6** | 0.5 |
| MetaShift (Dog-Indoor) | 0.8 | 0.8 | **0.8** |
| NIH (Pneumothorax-w/o tube) | 0.6 | 0.6 | **0.9** |
| RSNA (Cancer-w/o calcification) | 0.4 | 0.4 | **0.6** |
| VinDr (Cancer-w/o calcification) | 0.3 | 0.4 | **0.7** |

or tagging models, which struggle with missing or insufficient attributes, LADDER leverages LLM-driven latent medical knowledge to generate comprehensive hypotheses, enabling the discovery of contextual biases (subtypes or relationships) deeply embedded in the data. For waterbird classification (Fig.5c), LADDER retrieves sentences highlighting diverse ground truth water-related biases, *e.g.,* `boat`, `lake`. Next, LADDER

uses LLMs to analyze the sentences, generating hypotheses and prompts to test for biased attributes. The similarity score (Eq.1) tests these hypotheses to validate whether the absence of the attributes linked to each hypothesis results in a decline in classifier performance (Sec.2.2). For *e.g.,* in the waterbirds classification (Fig.5d), birds `sitting or flying` achieve 97.3% accuracy, while those not `sitting or flying` achieve 68.6%. In NIH (Fig.5b), pneumothorax patients with and without `chest tubes` yield accuracies of 97.7% and 48.1%, respectively. Refer to Appendix A.13.6, A.13.4, A.13.9, A.13.3 and A.13.8 for 1) more qualitative results, 2) the hypotheses closest to the ground truth biases, 3) the influence of different biased attributes via CLIP score, 4) impact of VLR pretraining datasets on bias detection in medical images, and 5) zero-shot classification improvements using extracted attributes, respectively.

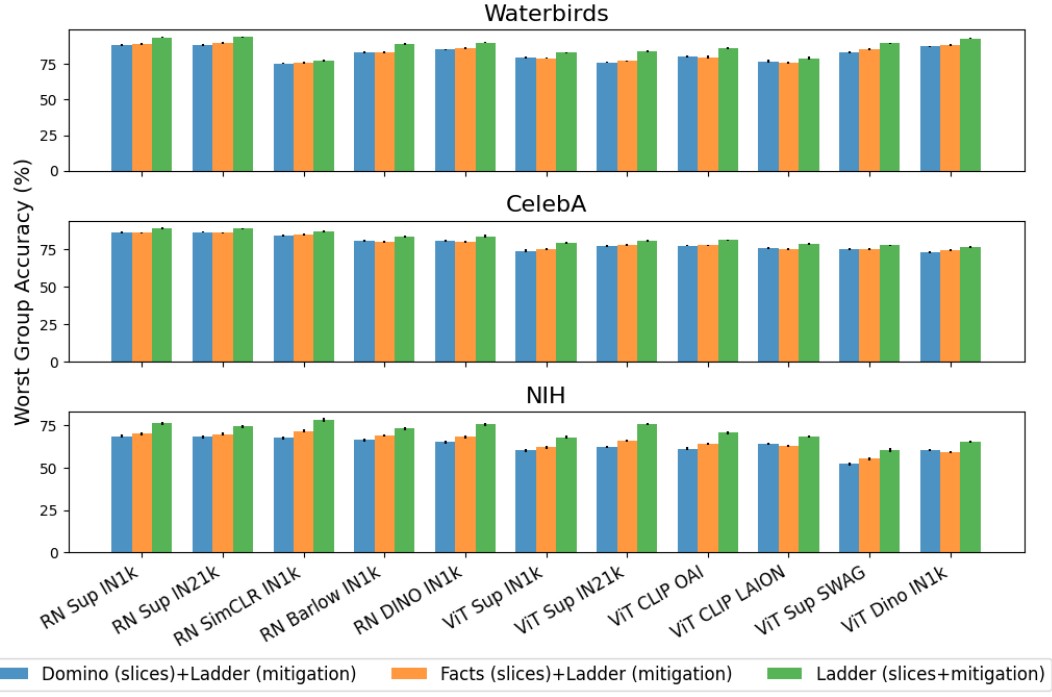

Figure 3: LADDER slices consistently outperform those from Domino and Facts when combined with LADDER's bias mitigation strategy across various settings.

**Attributes in the discovered hypotheses by LADDER across architectures and pre-taining methods (RQ3).** Yang et al. (2023) shows that every ERM-trained classifier ($f$) exhibits low WGA irrespective of architecture or pretraining due to consistently learning similar biases. Figure 4 illustrates that LADDER, leveraging LLM-driven reasoning and *domain knowledge*, consistently identifies similar biased attributes across different model architectures, pretraining methods, and datasets. In the NIH dataset, LADDER identifies key attributes such as `chest tubes`, `fluid levels` etc.across most classifiers. Also, in the Waterbirds dataset, LADDER detects attributes *e.g.,* `ocean` and `bamboo forest` consistently, highlighting the correlation of the spurious backgrounds with class labels and resembling the ground truth biases. Refer to Appendix A.13.11 for additional results.

**Benchmarking various bias mitigation algorithms (RQ4)** Tab. 4 shows that LADDER outperforms other bias mitigation baselines in estimating WGA, even without requiring the expensive ground truth shortcut attributes, for both training and validation datasets across CNN models (EN-B5 for Mammograms and RN Sup IN1k for

the rest). LADDER$_{bal}$ achieves a WGA of 93.7% and 76.4%, denoting a 6.2% and 7.3% improvement (↑) over DFR$_{val}$ in the Waterbirds and RSNA-Mammo datasets, respectively. For NIH, LADDER$_{bal}$ outperforms JTT and DFR$_{val}$ by 8.2% and 7.4%, respectively. Appendix A.13.7 reports the same for ViT-based models. Fig. 6 shows LADDER's consistent performance gain across various architectures and pre-training strategies. Tab. 11 in Appendix A.13.10 also shows that LADDER outperforms Li et al. (2023b) on multi-shortcut benchmark UrbanCars. Leveraging LLMs' advanced reasoning, LADDER accurately derives pseudo labels for the biased attributes from hypotheses to pinpoint true model biases. LADDER then applies targeted mitigation to address these biases by fine-tuning the last layer, resulting in a systematic debiased model per hypothesis. This efficient strategy effectively enhances model performance across the biases, modalities, and architectures.

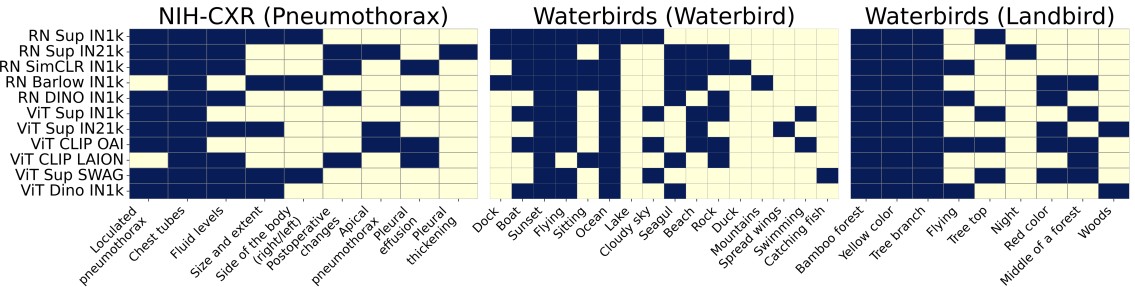

Figure 4: Biased attributes discovered in the hypotheses by LADDER across different architectures and datasets. LADDER discovers consistent biases irrespective of architectures and pertaining. Bright colors indicate attributes in LADDER's hypotheses, while light colors indicate their absence.

## 5 RELATED WORK

**Slice discovery.** Early slice discovery methods focus on tabular data to define slices (*e.g.,* nationality = Indian) (Chen et al., 2021; Chung et al., 2019; Sagadeeva & Boehm, 2021), struggling with unstructured data(*e.g.,* image or audio) due to the lack of coherent structure. Initial approaches (d'Eon et al., 2022; Sohoni et al., 2020; Kim et al., 2019; Singla et al., 2021) on unstructured data utilize clustering or dimensionality reduction to identify error slices. However, they lack comprehensive evaluation or qualitative analysis. Recent slice discovery methods include the usage of VLR space (Eyuboglu et al., 2022; Jain et al., 2022; Yenamandra et al., 2023; Zhang et al., 2023). For *e.g.,* Domino (Eyuboglu et al., 2022) projects data into

Table 5: Token usage and cost for each LLM. Each row shows the breakdown for an LLM extracting hypotheses across all 6 datasets, using RN Sup IN1k (natural images / CXRs) and EN-B5 (mammograms).

| Model Name | Input Tokens | Output Tokens | Total Cost |
|---|---|---|---|
| GPT-4o | 33,217 | 4,284 | $2.51 |
| Claude 3.5 Sonnet | 34,888 | 4,473 | $0.17 |
| Gemini 1.5 Pro | 33,872 | 4,378 | $0.32 |
| Llama 3.1 70B | 32,688 | 4,176 | $0.05 |
| **Total** | 134,665 | 17,311 | **$3.05** |

VLR space, identifies slices via a mixture model, and captions them. Facts(Yenamandra et al., 2023) amplifies spurious correlations in the initial training phase by increasing weight decay and discovering slices in VLR space. Both approaches compromise visual semantics, resulting in attribute inconsistencies within slices. DrML (Zhang et al., 2023) probes only CLIP-based classifiers using modality gap geometry and user-defined prompts, introducing subjectivity and potential human biases. Also, Facts and DrML are restricted to specific training setups, limiting generalizability to standard ERM classifiers. PRIME (Rezaei et al., 2023) uses expensive tagging models to discover attributes for slice discovery. HiBug (Chen et al., 2024) prompts LLMs (*e.g.,* ChatGPT) to suggest potentially biased attributes for error slices without any textual context from the data. Thus, it results in superficial keyword-based attributes derived purely from general user prompts, lacking

**NIH dataset for the pneumothorax class**

**(a) Sentences indicating the biased attributes**

1. perhaps mild **increase** in **hydropneumothorax** but with **chest tube** remaining in place and no striking change
2. in comparison with the study of ___ , there is little change in the **3 left chest tubes** with area of **hydro pneumothorax** persisting in the lateral aspect of the **upper left chest** as well as probably the left lung base
3. a **moderate sized loculated hydropneumothorax** shows decrease in **fluid** component and increasing gas component , particularly in the **right** base
4. small **right** pleural effusion has replaced the previous **basal pneumothorax** that developed with previous drainage of pleural effusion and placement of **2 thoracostomy tubes**
5. 2 right indwelling **pleural drains** are unchanged in their **respective positions** , and there has probably been some decrease in the volume of the **right posterior air** and pleural collection in the rib **right lower hemi thorax**
....

**(b) Performance of the model on slices where the biased attribute is present/absent**

**H1: loculated characteristics of pneumothorax:**

Present: 87.0 %
Absent: 49.0 %

**H2: presence of chest tubes:**

Present: 97.7 %
Absent: 48.1 %

**H3: Fluid levels in pneumothorax:**

Present: 88.2 %
Absent: 48.3 %

**H4: size and extent descriptions of pneumothorax:**

Present: 86.4 %
Absent: 49.8 %

**H5: side of the body affected by pneumothorax:**

Present: 87.2 %
Absent: 48.7 %

**Waterbirds dataset for the waterbird class**

**(c) Sentences indicating the biased attributes**

1. a seagul **sitting** on a dock with **boats** in the background
2. a duck swimming in the **ocean** with a blue sky and clouds in the background
3. a seagul **sits** on the **water** in front of a container **ship** at night
4. a seagul catching a fish in the **ocean**
5. a seagul **sitting** on a rock in the **ocean**
6. a seagul in the **water** with its wings spread out
7. a seagul **sitting** on a rock in front of a lighthouse at **sunset**
8. a seagul **flying** over the **water** with a fish in it's mouth
9. a seagul **flying** over the **ocean** at **sunset**
10. a seagul **flying** over the **ocean** with rocks and cliffs in the background
11. a seagul **sitting** on a rock in the middle of a **lake**
12. a seagul **flying** over the **beach** with a blue sky in the background
....

**(d) Performance of the model on slices where the biased attribute is present/absent**

**H1: Specific background elements like docks and boats:**

Present: 97.0 %
Absent: 68.8 %

**H2: Specific times of day like sunset:**

Present: 95.5 %
Absent: 70.1 %

**H3: Specific actions like flying or sitting:**

Present: 97.3 %
Absent: 68.6 %

**H4: Presence of water bodies like oceans and lakes:**

Present: 97.6 %
Absent: 68.2 %

**H5: Weather conditions like cloudy skies:**

Present: 95.3 %
Absent: 70.2 %

Figure 5: LADDER discovers slices for biased attributes in RN Sup IN1k classifier for *pneumothorax waterbird* classification in **NIH** and **Waterbirds** datasets respectively. Panels (a) and (c) show sentences retrieved by LADDER showing biased attributes present in correctly classified samples but missing in others. Panels (b) and (d) illustrate the model's performance when biased attributes are either present or absent. Hypotheses indicative of ground truth biases (*e.g.,* water for waterbirds) are highlighted in yellow.

the deeper contextual grounding needed for bias detection. B2T (Kim et al., 2024) extracts keywords from captions. All these methods face limitations due to the incompleteness of tags or keyword-based attributes. Moreover, none of these methods incorporate any reasoning or latent *domain knowledge*, which are critical for specialized fields, *e.g.,* radiology. **Bias mitigation.** Bias mitigation aims to enhance various subgroup performance. GroupDRO (Sagawa et al., 2020) targets high-error groups, while LfF (Nam et al., 2020) adjusts gradient contributions through biased and debiased model pairs. JTT (Liu et al., 2021) identifies and reweights minority groups, while DFR (Kirichenko et al., 2022) retrains the final layer using a balanced validation set. All these methods require group annotations and focus on mitigating errors in the worst-performing group, often amplifying errors in other subgroups. Spuriosity Rankings (Moayeri et al., 2023b) rank data samples based on spurious features but introduce subjectivity bias due to human-in-the-loop component. Li et al. (2023b) mitigates multiple biases using an ensemble-based approach but relies on predefined bias types, which limits its adaptability to unknown biases. LADDER overcomes all limitations in slice discovery and bias mitigation. For discovery, LADDER incorporates the *domain knowledge* of LLMs, reason about model errors, and generates hypotheses identifying nuanced biases from any pretrained model without external tags

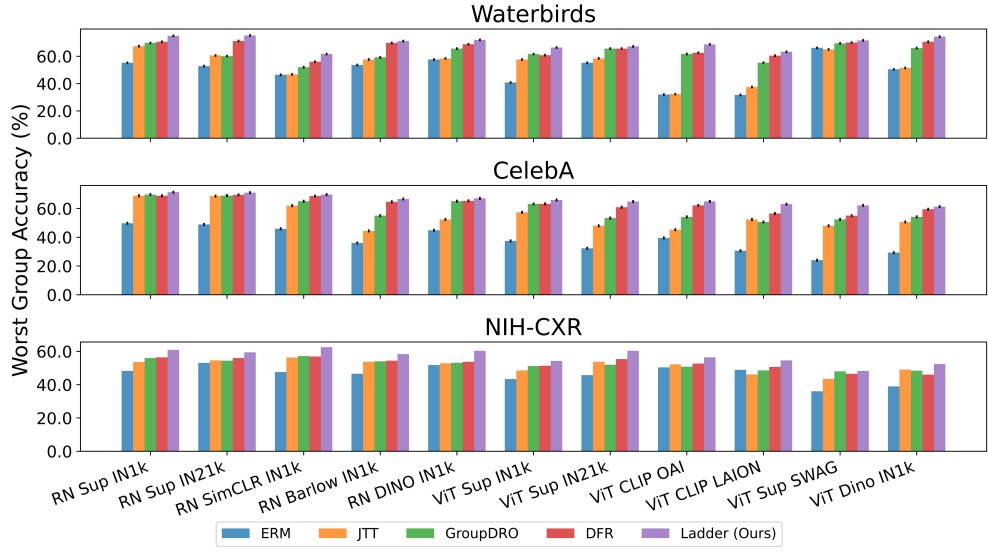

Figure 6: WGA across bias mitigation methods. LADDER consistently outperforms other bias mitigation baselines (ERM, JTT, GroupDRO, and DFR) across different model architectures and pre-training strategies.

or predefined attributes, unlike existing methods. Then, LADDER generates pseudo-labels for each bias, and fine-tunes the classifier's last layer cost-effectively, mitigating multiple biases automatically – without any group annotations, predefined bias types, or human intervention.

## 6 OVERALL COST AND CHOICE OF LLMS

Tab. 5 shows the cost of using various LLMs. Each row shows the total breakdown for an LLM extracting hypotheses across all 6 datasets, using RN Sup IN1k (natural images/CXRs) and EN-B5 (mammograms). For each dataset, LADDER invokes LLM once using sentences only (no images). The total cost incurred in this paper is ∼$28 across all architectures and pretraining used in the experiments. Thus, LLMs are far more cost-effective than developing new tagging models for unexplored domains *e.g.,* radiology, or manually annotating shortcuts for entire datasets. Fig. 31 in Appendix A.13.14 shows the attributes identified by each LLM while generating hypotheses. Different LLMs capture distinct sets of attributes, yet substantial overlap exists, with many attributes consistently revealing actual biases across models. Ablation studies in Appendix A.13.15 indicate that using different LLMs to compute WGA shows that Gemini and GPT-4o achieve higher WGA for medical images than the others.

## 7 CONCLUSION AND LIMITATION

This paper presents LADDER, a method to discover and mitigate error slices using natural language and LLM reasoning, without relying on costly external attribute annotations. LADDER generates hypotheses to detect model errors and mitigates biases using pseudo-labels tailored to each identified slice. Extensive experiments show LADDER's efficacy compared to baselines. However, its performance depends on the quality of available captions and the vision-language model. Future work will focus on iterative refinement of the discovery process based on slice complexity and leveraging language inversion to eliminate the need for a text corpus.

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

# A APPENDIX

CONTENTS

## A.1 GLOSSARY OF TERMS AND NOTATIONS

| | |
|---|---|
| **LLM** | Large Language Model |
| $\mathcal{X}$ | Set of input images. |
| **Y** | Set of labels. |
| $X_Y$ | Set of images belonging to class label $Y$. |
| $f$ | Trained classifier predicting class labels $\mathcal{Y}$ from images $\mathcal{X}$. |
| $g$ | Classification head of the model. |
| $\Phi$ | Image representation function of the classifier $f$. |
| $\Psi^I$, $\Psi^T$ | Image and text encoders in the joint vision-language representation (VLR) space. |
| $\langle \cdot, \cdot \rangle$ | Dot product operation, used to compute similarity scores between embeddings. |
| $\pi$ | Projection function mapping image representations $\Phi$ to the VLR space $\Psi^I$. |
| $\Delta^I$ | Difference in mean of the projected representations of correctly and incorrectly classified samples. |
| $t_{\mathbf{val}}$ | Validation text corpus, such as radiology reports or image captions. |
| $\mathbb{S}_Y$ | Error slices corresponding to class $Y$, *i.e.,* subset of images with class $Y$ where the classifier $f$ underperforms. |
| $\mathcal{S}_{Y,\neg\mathtt{attr}}$ | Ground truth error slice without attribute $\mathtt{attr}$ and class $Y$. |
| $e(\cdot)$ | Error rate function for a given subset of data. |
| **topK** | $\mathtt{topK}$ sentences having highest similarity with $\Delta^I$ |
| $\mathcal{H}$ | Set of hypotheses generated by LLM, each of which is an indicator of an attribute on which $f$ may be biased. |
| $\mathcal{T}$ | Set of sentences corresponding to test each of the set of hypotheses $H$. |
| $\mathcal{T}_H$ | Set of sentence for the hypothesis $H \in \mathcal{H}$ |
| $s_H$ | Similarity score for a hypothesis $H$, measuring alignment of image representations with hypothesis-specific attributes. |
| $\mathcal{S}_{\neg H,Y}$ | Subset of images for class $Y$ not aligning with the hypothesis $H$, a potential candidate for an error slice. |
| $\mathscr{R}$ | Retrieval function for selecting sentences with high similarity to $\Delta I$. |
| $\tau$ | Threshold value for selecting images based on similarity scores for error slice identification. |

## A.2 Learning Projection ($\pi$) from classifier to VLR space

$\pi$ is a learnable projection function, $\pi : \Phi \to \Psi^I$, projecting the image representation of the classifier $\Phi(x)$ to the VLR space, $\Psi(x)$, where $x \in \mathcal{D}_{train}$. $\mathcal{D}_{train}$ denotes the training set. We follow Moayeri et al. (2023a) to learn $\pi$. Specifically, $\pi$ is an affine transformation, *i.e.*, $\pi_{W,b}(z) = W^T z + b$, where $W$ and $b$ are the learnable weights and biases of the projector $\pi$. To retain the original semantics in the classifier representation space, we optimize the following objective:

$$W, b = \arg\min_{W,b} \frac{1}{\mathcal{D}_{train}} \sum_{x \in \mathcal{D}_{train}} \|W^T \Phi(x) + b - \Psi(x)\|_2^2 \qquad (2)$$

## A.3 Precision@k

**Precision@k** Eyuboglu et al. (2022); Yenamandra et al. (2023) measures the degree to which the predicted slices overlap with the ground truth slices in a dataset.

Let $S = \{s_1, s_2, \ldots, s_l\}$ represent the ground truth bias-conflicting slices in a dataset $\mathcal{D}$. A slice discovery algorithm $A$ predicts a set of slices $\hat{S} = \{\hat{s}_1, \hat{s}_2, \ldots, \hat{s}_m\}$. For each predicted slice $\hat{s}_j$, let $O_j = \{o_{j1}, o_{j2}, \ldots, o_{jn}\}$ denote the sequence of sample indices ordered by the decreasing likelihood that each sample belongs to the predicted slice $\hat{s}_j$.

Given a ground truth slice $s_i$ and a predicted slice $\hat{s}_j$, we compute their similarity as:

$$P_k(s_i, \hat{s}_j) = \frac{1}{k} \sum_{i=1}^{k} \mathbb{I}[x_{o_{ji}} \in s_i],$$

where $P_k(s_i, \hat{s}_j)$ is the proportion of the top $k$ samples in the predicted slice $\hat{s}_j$ that overlap with the samples in the ground truth slice $s_i$, and $\mathbb{I}$ is an indicator function that returns 1 if the sample belongs to $s_i$ and 0 otherwise.

For each ground truth slice $s_i$, we map it to the most similar predicted slice $\hat{s}_j$ by maximizing $P_k(s_i, \hat{s}_j)$. We then compute the average similarity score between the ground truth slices and their best-matching predicted slices. Specifically, the `Precision@k` for a slice discovery algorithm $A$ is given by:

$$\texttt{Precision@k}(A) = \frac{1}{l} \sum_{i=1}^{l} \max_{j \in [m]} P_k(s_i, \hat{s}_j),$$

where $l$ is the number of ground truth slices, $m$ is the number of predicted slices, and $P_k(s_i, \hat{s}_j)$ is the similarity score for the ground truth slice $s_i$ and predicted slice $\hat{s}_j$.

This metric evaluates how well the algorithm's predicted slices match the bias-conflicting slices in the dataset, with higher scores indicating better alignment between predicted and ground truth slices. By computing the `Precision@k`, we can assess the effectiveness of slice discovery algorithms in identifying and isolating the most significant bias-conflicting regions in the data.

## A.4 AccGAP

This metric quantifies the absolute difference in accuracy between the ground truth error slice (w/o a specific attribute) and the predicted slice (the subset of data that doesn't follow the top 3 closest matching hypotheses).

$$\texttt{AccGap} = \left| \frac{1}{|\mathcal{S}_{Y, \neg \texttt{attr}}|} \sum_{X \in \mathcal{S}_{Y, \neg \texttt{attr}}} \mathbf{1}\{f(X) = Y\} - \frac{1}{3} \sum_{i=1}^{3} \left( \frac{1}{|\mathcal{S}_{Y, \neg H_i^*}|} \sum_{X \in \mathcal{S}_{Y, \neg H_i^*}} \mathbf{1}\{f(X) = Y\} \right) \right|, \quad (3)$$

where $\mathcal{S}_{Y,\neg\texttt{attr}}$ is the ground truth error slice with class $Y$ and w/o the attribute $\texttt{attr}$; $\mathcal{S}_{Y,\neg H_i^*}$ is the predicted error slice with class $Y$ and not following the hypothesis $H_i^*$. We compute the top 3 hypotheses $\{H_i^*\}_{i=1}^3$, closest to the attribute $\texttt{attr}$ using:

$$H^* = \arg\max_{H \in \mathcal{H}}(\langle \Psi^T(\mathcal{T}_H), \Psi^T(\texttt{attr})\rangle), \tag{4}$$

For Domino and FACTS, we use the $\texttt{top5}$ captions per slice to discover the $\texttt{top3}$ error slices using Eq. 4 and finally compute **AccGap**. Refer to Appendix A.4 for the results.

## A.5 CLIP SCORE

Kim et al. (2024) introduces the CLIP score, a metric that leverages the similarity between language and vision embeddings to quantify the influence of specific attributes on misclassified samples. In their method, attributes frequently present in misclassified images receive a high CLIP score, while absent ones score lower. For instance, in the Waterbirds dataset, the CLIP score for "bamboo" is high, as many misclassified waterbirds appear with bamboo in the background.

We propose a modification to the CLIP score. As discussed in Sec. 2.1, our goal is to identify visual attributes that are prevalent in correctly classified samples but absent in misclassified ones. This approach provides deeper insights into the attributes contributing to correct classifications, which is particularly valuable for medical images. In scenarios such as pneumothorax detection in the NIH dataset, understanding biases incorrectly classified cases—such as the presence of chest tubes—can help isolate features that lead to reliable diagnoses while addressing spurious correlations. Formally we define the CLIP score corresponding to the attribute $\texttt{attr}$ and a dataset $\mathcal{D}$ as,

$$s_{CLIP}(\texttt{attr}, \mathcal{D}) = \texttt{sim}(\texttt{attr}, \mathcal{D}_{correct}) - \texttt{sim}(\texttt{attr}, \mathcal{D}_{wrong}),$$

where $\texttt{attr}$ is the attribute obtained from the specific hypothesis by LLM, described in Sec. 2.2, $\mathcal{D}_{correct}$ and $\mathcal{D}_{wrong}$ are the correctly classified and misclassified samples. Also, $\texttt{sim}(\texttt{attr}, \mathcal{D})$ is the similarity between the attribute $\texttt{attr}$ and the dataset $\mathcal{D}$, estimated as the average cosine similarity between normalized embedding of a word $\Psi^T(\texttt{attr})$ and images $\Psi^I(\texttt{x})$ for $x \in \mathcal{D}$, where

$$\texttt{sim}(\texttt{attr}, \mathcal{D}) = \frac{1}{\mathcal{D}} \sum_{x \in \mathcal{D}} \Psi^I(x)\Psi^T(\texttt{attr})$$

Refer to Appendix A.13.9 for the results.

## A.6 PROMPTS USED BY LLM FOR HYPOTHESES GENERATION DISCUSSED IN SEC. 2.2

The following is a general template of the prompt utilized to generate the hypotheses from LLM, discussed in Sec. 2.2. In this template, we substitute the <task> placeholders with bird species, hair color, animal species, pneumothorax, cancer, and abnormality based on the corresponding dataset – Waterbirds, CelebA, MetaShift, NIH, RSNA-Mammo, and VinDr-Mammo. The modalities are natural images, chest-x-rays, and 2D mammograms. **Crucially, we only replace these two placeholders. We never include the actual dataset names or words like "water", "land", "gender", "tube", "background" or any other attributes leading to model's mistakes in the prompt, as these could bias the LLM's output**. For medical images, we also add: Ignore '___' as they are due to anonymization. We focus only on positive <disease> patients, as many reports consist of '___' for clarity. top <K> depends on the dataset discussed in the experiment section (Sec. 3).

## Prompt for hypothesis generation

**Context:** <task> classification from <modality> using a deep neural network
**Analysis post-training:** On a validation set,

    a. Get the difference between the image embeddings of correct and incorrectly classified samples to estimate the features present in the correctly classified samples but missing in the misclassified samples.

    b. Retrieve the top <K> sentences from the radiology report that match closely to the embedding difference in step a.

    c. The sentence list is given below:

> **`topK` sentence list (retrieved using Sec. 2.1)**

These sentences represent the features present in the correctly classified samples but missing in the misclassified samples.
**Task:**
Consider the consistent attributes present in the descriptions of correctly classified and misclassified samples regarding <task>. Formulate hypotheses based on these attributes. Attributes are all the concepts (*e.g.,* explicit or implicit anatomies, observations, any symptom of change related to the disease, or any concept leading to potential bias in medical images or any visual cues present in the natural images) in the sentences. Assess how these characteristics might be influencing the classifier's performance. Your response should contain only the list of top hypotheses and nothing else. For the response, you should use the following Python dictionary template with no extra sentences:

```
hypothesis_dict =
{
'H1': 'The classifier is making mistake as it is biased toward <attribute>',
'H2': 'The classifier is making mistake as it is biased toward <attribute>',
'H3': 'The classifier is making mistake as it is biased toward <attribute>',
    ...
}
```

To effectively test Hypothesis1 (H1) using the CLIP language encoder, you must create prompts that explicitly validate H1. These prompts will help to generate text embeddings that capture the essence of the hypothesis, which can be used to compute similarity with the image embeddings from the dataset. The goal is to see if the images where the model makes mistakes align with H1 or violate H1. The prompts are a Python list. Remember, your focus is only on the "<task>".
Do this for all the hypotheses. Your final response should follow the following list of dictionaries, nothing else:

```
prompt_dict = {
'H1_<attribute>': [List of prompts],
'H2_<attribute>': [List of prompts],
    ...
}
```

Each attribute hypothesis should contain 5 prompts. So the final response should follow the below format strictly (nothing else, no extra sentence):
```python

hypothesis_dict
prompt_dict
```

## A.7 EXTENDED DETAILS ON DATASETS

### WATERBIRDS

The **Waterbirds** dataset (Wah et al., 2011) is frequently employed in studies addressing spurious correlations. This binary classification dataset overlaps images from the Caltech-UCSD Birds-200-2011 (CUB) dataset with backgrounds sourced from the Places dataset (Zhou et al., 2017). The primary task involves determining whether a bird depicted in an image is a landbird or a waterbird, with the background (water or land) as the spurious attribute. For consistency and comparability, we adhere to the train/validation/test splits utilized in prior research (Guo et al., 2020).

### CELEBA

The **CelebA** dataset (Liu et al., 2015) comprises over 200,000 images of celebrity faces. In the context of spurious correlations research, this dataset is typically used for the binary classification task of predicting hair color (blond vs. non-blond), with gender serving as the spurious correlation. In alignment with previous studies (Guo et al., 2020), we use the standard dataset splits. The CelebA dataset is available under the Creative Commons Attribution 4.0 International license.

### METASHIFT

The **MetaShift** dataset (Liang & Zou, 2022) offers a flexible platform for generating image datasets based on the Visual Genome project (Krishna et al., 2017). Our experiments utilize the pre-processed *Cat vs. Dog* dataset, designed to differentiate between cats and dogs. The dataset features the image background as a spurious attribute, with cats typically appearing indoors and dogs outdoors. We use the "unmixed" version of this dataset, as provided by the authors' codebase.

### NIH CHESTXRAYS

The **NIH** ChestX-ray dataset (Wang et al., 2017), also known as ChestX-ray14, is a large dataset of chest radiographs (X-rays) provided by the National Institutes of Health (NIH). The dataset comprises 112,120 frontal-view X-ray images of 30,805 unique patients. Each image is associated with one or more of the 14 labeled thoracic diseases, which include atelectasis, cardiomegaly, effusion, infiltration, mass, nodule, pneumonia, pneumothorax, consolidation, edema, emphysema, fibrosis, pleural thickening, and hernia. Previous works (Docquier & Rapoport, 2012) show that most pneumothorax patients have a spurious correlation with the chest drains. Chest drains are used to treat positive Pneumothorax cases. We adopt the strategy discussed in Murali et al. (2023) to annotate ches drains for each sample. We use the official train/val/test split (Wang et al., 2017).

### RSNA BREAST MAMMOGRAMS

The **RSNA-Mammo** dataset[2] is a publicly available dataset containing 2D mammograms from 11,913 patients, with 486 diagnosed cancer cases. The task is to classify malignant cases from screening mammograms. We use a 70/20/10 train/validation/test split for evaluation as (Ghosh et al., 2024).

### VINDR BREAST MAMMOGRAMS

The **VinDr-Mammo** dataset[3] (Nguyen et al., 2023) is a publicly available 2D mammogram dataset of 5,000 exams (20,000 images) from Vietnam, each with four views. It includes breast-level BI-RADS assessment

---

[2]https://www.kaggle.com/competitions/rsna-breast-cancer-detection
[3]https://www.physionet.org/content/vindr-mammo/1.0.0/

categories (1-5), breast density categories (A-D), and annotations for mammographic attributes (e.g., mass, calcifications). Following (Wen et al., 2024), we classify patients with BI-RADS scores between 1 and 3 as normal and those with scores of 4 and 5 as abnormal. We adopt the train-test split from (Nguyen et al., 2023).

## A.8 EXTENDED DETAILS ON SLICE DISCOVERY ALGORITHMS

**Domino.** Domino Eyuboglu et al. (2022) identifies systematic errors in machine learning models by leveraging cross-modal embeddings. It operates in three main steps: embedding, slicing, and describing.

1. **Embedding**: Domino uses cross-modal models (*e.g.,* CLIP) to embed inputs and text in the same latent space. This enables the incorporation of semantic meaning from text into input embeddings, which is crucial for identifying coherent slices.

2. **Slicing**: It employs an error-aware mixture model to detect underperforming regions within the embedding space. This model clusters the data based on embeddings, class labels, and model predictions to pinpoint areas where the model performance is subpar. The mixture model ensures that identified slices are coherent and relevant to model errors.

3. **Describing**: Domino generates natural language descriptions for the discovered slices. It creates prototype embeddings for each slice and matches them with text embeddings to describe the common characteristics of the slice. This step provides interpretable insights into why the model fails on these slices.

Domino's approach improves slice coherence and generates meaningful slice descriptions.

**Facts.** Facts Yenamandra et al. (2023) (First Amplify Correlations and Then Slice) aims to identify bias-conflicting slices in datasets through a two-stage process:

1. **Amplify Correlations**: This stage involves training a model with a high regularization term to amplify its reliance on spurious correlations present in the dataset. This step helps segregate biased-aligned from bias-conflicting samples by making the model fit a simpler, biased-aligned hypothesis.

2. **Correlation-aware Slicing**: In this stage, FACTS uses clustering techniques on the bias-amplified feature space to discover bias-conflicting slices. The method identifies subgroups where the spurious correlations do not hold, highlighting areas where the model underperforms due to these biases.

Facts leverages a combination of bias amplification and clustering to reveal underperforming data slices, providing a foundation for understanding and mitigating systematic biases in machine learning models.

## A.9 EXTENDED DETAILS ON ERROR MITIGATION BASELINES

We categorize the various bias mitigation algorithms and provide detailed descriptions for each category below.

### VANILLA

The empirical risk minimization (ERM) algorithm, introduced by Vapnik (Vapnik, 1999), seeks to minimize the cumulative error across all samples.

### SUBGROUP ROBUST METHODS

**GroupDRO:** GroupDRO (Sagawa et al., 2020) propose Group Distributionally Robust Optimization (Group-DRO), which enhances ERM by prioritizing groups with higher error rates. **CVaRDRO:** Duchi and Namkoong

(Duchi & Namkoong, 2021) introduce a variant of GroupDRO that dynamically assigns weights to data samples with the highest losses. **LfF:** LfF (Nam et al., 2020) concurrently trains two models: the first model is biased, and the second is de-biased by re-weighting the loss gradient. **Just Train Twice (JTT):** JTT (Liu et al., 2021) propose an approach that initially trains an ERM model to identify minority groups in the training set, followed by a second ERM model where the identified samples are re-weighted. **LISA:** LISA (Yao et al., 2022) utilizes invariant predictors through data interpolation within and across attributes. **Deep Feature Re-weighting (DFR):** DFR (Kirichenko et al., 2022) suggests first training an ERM model and then retraining the final layer using a balanced validation set with group annotations.

### DATA AUGMENTATION

**Mixup:** Mixup (Zhang et al., 2018) proposes an approach that performs ERM on linear interpolations of randomly sampled training examples and their corresponding labels.

### DOMAIN-INVARIANT REPRESENTATION LEARNING

**Invariant Risk Minimization (IRM):** IRM (Arjovsky et al., 2020) learns a feature representation such that the optimal linear classifier on this representation is consistent across different domains. **Maximum Mean Discrepancy (MMD):** MMD (Li et al., 2018) aims to match feature distributions across domains. **Note: All methods in this category necessitate group annotations during training**.

### IMBALANCED LEARNING

**Focal Loss (Focal):** Focal (Lin et al., 2017) introduces Focal Loss, which reduces the loss for well-classified samples and emphasizes difficult samples. **Class-Balanced Loss (CBLoss):** CBLoss (Cui et al., 2019) suggests re-weighting by the inverse effective number of samples. **LDAM Loss (LDAM):** LDAM (Cao et al., 2019) employs a modified margin loss that preferentially weights minority samples. **Classifier Re-training (CRT):** CRT (Kang et al., 2020) decomposes representation learning and classifier training into two distinct stages, re-weighting the classifier using class-balanced sampling during the second stage. **ReWeightCRT:** ReWeightCRT (Kang et al., 2020) proposes a re-weighted variant of CRT.

### A.10 EXTENDED DETAILS ON EXPERIMENTS

#### A.10.1 IMPLEMENTATION DETAILS OF THE SOURCE MODEL $f$ USING ERM

For natural images and chest X-rays (CXRs), we resize the images to 224×224 and train ResNet-50 (RN)(He et al., 2016) and Vision Transformer (ViT)(Dosovitskiy et al., 2020) models as $f$ to predict labels. We explore various pretraining methods for initializing model weights, including supervised learning (Sup), SIMCLR(Chen et al., 2020), Barlow Twins (Zbontar et al., 2021), DINO (Caron et al., 2021), and CLIP-based pretraining (Radford et al., 2021). The pretraining datasets utilized include ImageNet-1K (IN1)(Deng et al., 2009), ImageNet-21K (IN-21K)(Ridnik et al., 2021), SWAG (Singh et al., 2022), LAION-2B (Schuhmann et al., 2022), and OpenAI-CLIP (OAI) (Radford et al., 2021). For instance,"RN Sup IN1k" refers to a ResNet model pretrained using supervised learning and ImageNet-1K.

We train both ResNet and ViT models as $f$ for natural images and NIH-CXR following the setup in Yang et al. (2023)[4]. Preprocessing steps include resizing the images to 224×224, applying center-cropping, and normalizing the images using ImageNet channel statistics. Consistent with prior work (Guo et al., 2020; 2019), we apply stochastic gradient descent (SGD) with momentum for optimization across all image datasets. Each model is trained for a total of 30,000 steps across all datasets, with specific training on Waterbirds and

---

[4]https://github.com/YyzHarry/SubpopBench

MetaShift for 5,000 steps each. For NIH, we utilize the Adam optimizer with a learning rate of 0.0001 and train for 60 epochs to achieve optimal convergence.

For RSNA-Mammo, we leverage the setting from one of the leading Kaggle competition solutions[5]. In this setup, the images are resized to 1520×912, and we train an EfficientNet-B5 model (Tan & Le, 2019) for 9 epochs using the SGD optimizer, with a learning rate of 5e-5 and a weight decay of 1e-4.

Additionally, for CXR-CLIP, we use their pretrained models[6], which were trained on MIMIC-CXR and CheXpert datasets. For Mammo-CLIP, we utilize their EN-B5 variant[7].

### A.10.2 Ablations

For the captioning ablations, we compare the performance of LADDER using BLIP (Li et al., 2022), BLIP-2 (Li et al., 2023a), ClipCap (Mokady et al., 2021), and GPT-4o (Wu et al., 2024). Additionally, for LLMs, we compare the performance of LADDER with GPT-4o (Wu et al., 2024), Claude 3.5 Sonnet, Llama 3.1 70B (Dubey et al., 2024), and Gemini 1.5 Pro (Team et al., 2024).

### A.10.3 Radiology text synthesis for 2D Mammograms

In Ghosh et al. (2024), the authors generate mammography reports using labeled mammographic attributes from the VinDr dataset in collaboration with a board-certified radiologist. This approach leverages the templated nature of breast mammogram reports, which are more standardized than those for other medical imaging modalities. This standardized structure follows protocols like BI-RADS (Breast Imaging-Reporting and Data System), which promotes uniformity in reporting (Palanisamy et al., 2023). Specifically, they focus on the following attributes: `mass`, `architectural distortion`, `calcification`, `asymmetry (focal, global)`, `density`, `suspicious lymph nodes`, `nipple retraction`, `skin retraction`, and `skin thickening`. Then they follow the report templates with radiologist-defined prompts in Ghosh et al. (2024), describing key parameters such as:

- **Attribute Value**: Positive, negative, etc.
- **Subtype**: Suspicious, obscured, spiculated, etc.
- **Laterality**: Left or right breast.
- **Position**: Upper, lower, inner, outer quadrant.
- **Depth**: Anterior, mid, or posterior.

Finally, they generate concise report-like sentences by substituting these values into the templates. The authors leverage these sentences in Mammo-FActOR to perform weakly supervised localization of mammographic findings. In our work, we collect all these sentences to probe the EN-B5 classifier $f$, analyzing its errors during the retrieval step (Sec. 2.1) for the RSNA-Mammo and VinDr-Mammo datasets.

Below are some examples of mammography report sentences corresponding to the specific mammographic attributes.

**Mass:**

```
    "there is a mass in the right breast",
    "there is a mass in the right breast at anterior depth",
```

---

[5] https://github.com/Masaaaato/RSNABreast7thPlace
[6] https://github.com/kakaobrain/cxr-clip
[7] https://huggingface.co/shawn24/Mammo-CLIP/blob/main/Pre-trained-checkpoints/
b5-model-best-epoch-7.tar

```
"there is a mass in the upper right breast at mid-depth."
...
```

**Architectural distortion:**

```
"there is architectural distortion in the right breast",
"there is architectural distortion in the right breast at anterior depth",
"there is architectural distortion in the right breast at mid-depth",
...
```

**Calcification:**

```
"there is calcification in the right breast",
"there is calcification in the right breast at anterior depth",
"there is calcification in the right breast at mid depth",
...
```

**Asymmetry:**

```
"there is a developing asymmetry in the outer right breast",
"there is an asymmetry in the inner right breast at anterior depth",
"there is an asymmetry in the right breast at mid-depth"",
...
```

**Global Asymmetry:**

```
"there is a global asymmetry in the right breast",
"there is a new global asymmetry in the right breast",
"there is a global asymmetry in the inner right breast"
...
```

**Focal Asymmetry:**

```
"there is a focal asymmetry in the right breast",
"there is a focal asymmetry in the right breast at anterior depth",
"there is a focal asymmetry in the right breast at mid depth",
...
```

**Density:**

```
"the breasts being almost entirely fatty",
"scattered areas of fibroglandular density",
"the breast tissue is heterogeneously dense",
"the breasts are extremely dense"
```

**Suspicious lymph node:**

```
"there is a suspicious lymph node in the  right axilla",
"there is a hyperdense lymph node in the  right axillary tail",
"there is an increased lymph node in the  right axillary tail",
...
```

**Suspicious lymph node:**

```
    "there is a suspicious lymph node in the  right axilla",
    "there is a hyperdense lymph node in the  right axillary tail",
    "there is an increased lymph node in the  right axillary tail",
    ...
```

**Nipple retraction:**

```
    "there is a new nipple retraction in the right breast",
    "there is an increased nipple retraction in the right breast",
    "there is a possible nipple retraction in the right breast",
    ...
```

**Skin retraction:**

```
    "there is  skin retraction in the right breast",
    "there is  skin retraction in the inner right breast",
    "there is  skin retraction in the lower right breast",
    ...
```

**Skin thickening:**

```
    "there is increasing skin thickening of the periareolar right breast",
    "there is asymmetric skin thickening of the lower right breast",
    "there is asymmetric skin thickening of the inner right breast",
    ...
```

A.11    TOY DATASET CONSTRUCTION

We construct a synthetic dataset based on the **CUB-200-2011** (Wah et al., 2011) dataset, classifying bird species into two categories: **Class 0** ($y = 0$) and **Class 1** ($y = 1$). Class 1 consists of the following bird species: *Albatross*, *Auklet*, *Cormorant*, *Frigatebird*, *Fulmar*, *Gull*, *Jaeger*, *Kittiwake*, *Pelican*, *Puffin*, *Tern*, *Gadwall*, *Grebe*, *Mallard*, *Merganser*, *Guillemot*, and *Pacific Loon*. All remaining bird species are assigned to Class 0. To introduce spurious correlations, we overlay two 3D boxes on each image. In the training set for Class 0, the majority of samples (95%) were biased, with the yellow box consistently placed to the left of the red box. For Class 1, the boxes were randomly placed, introducing variability in their positioning. In the validation and test sets, we split the positioning evenly, with 50% biased and 50% random samples across both classes, ensuring a balanced evaluation of the model's reliance on spurious cues.

The primary goal of this dataset is to introduce a form of *reasoning* beyond the mere presence or absence of spurious correlations. Unlike prior datasets that rely on background cues (*e.g.,* Waterbirds or Metashift) or attributes like gender (*e.g.,* CelebA), our dataset integrates positional reasoning. Specifically, for Class 0, the yellow box is consistently placed to the left of the red box, creating a spurious correlation. For Class 1, the boxes are randomly positioned, removing this shortcut. The relative positioning of the boxes allows the captions to encode spatial relationships, which can be consumed by large language models (LLMs) to reason about these spatial cues. We train an ImageNet pretrained-ResNet model (RN Sup IN1k) on this dataset. Predictably, the classifier latches onto the spurious correlation of rectangle position, leading to underperformance on subsets where the shortcut is absent. The model achieves a mean accuracy of 85.6% and a worst-group accuracy (WGA) of 65.2%.

To analyze the model's errors, we generate a corpus of rich captions for the validation set using a GPT-4o-based captioner. These captions describe both the presence of the rectangle and its position relative to the bird. Using LADDER, we aim to detect the reason for the classifier's mistakes and mitigate it. LADDER leverages

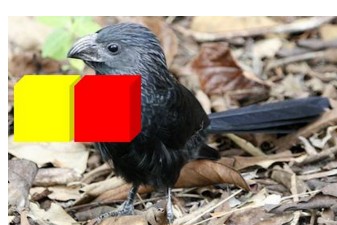

**Extracted hypotheses by Ladder**
The classifier is making mistake as it is biased toward:
**H1:** relative positioning of red and yellow box
**H2:** images with small birds
**H3:** images with overlapping boxes
**H4:** the position of boxes relative to the bird
**H5:** images with bird on branches

Figure 7: Sample images of our toy dataset to validate the reasoning of LLM utilized by LADDER. The dataset has two classes. Images with class 0 are biased, with the yellow box always placed to the left of the red box. For images with class 1, the boxes are randomly placed.

the reasoning capabilities of LLMs to capture both the presence of the rectangles and their relative spatial position. In contrast, methods *e.g.,* PRIME, rely on external tagging models, which only detect the presence or absence of shortcuts. Furthermore, since LADDER discovers biased attributes via LLM-generated reasoning, it can effectively mitigate these biases without requiring ground truth annotations or prior knowledge of the attributes.

The data is split into training, validation, and test sets, with all metadata (including labels, rectangle positions) saved for future analysis.

## A.12 COMPUTING RESOURCES

All the models are implemented in PyTorch and trained on a single NVIDIA RTX-6000 GPU with 32G of memory. We use the checkpoints from the official repository of CLIP and CXR-CLIP. Classifiers trained with natural images take 2-3 hours to train. The Classifier with NIH-CXR takes approximately 4-6 hours to train. The Classifier with RSNA-Mammo and VinDr-Mammo takes approximately 21 and 15 hours to train.

## A.13 EXTENDED RESULTS

### A.13.1 LANGUAGE AS AN ALTERNATIVE TO ATTRIBUTES TO ANALYZE THE ERRORS

In this experiment, we ask this fundamental question, "can language be used as an alternative to discover the slices?" As illustrated in Tab. 6, utilizing caption embeddings instead of explicit attributes achieves comparable RMSE for predicting model errors on CelebA and Waterbirds datasets. The strong Spearman and Pearson correlation coefficients between model errors predicted by language embeddings and biased attributes suggest that language effectively captures bias patterns, providing a reliable proxy for ground truth attributes.

Table 6: Comparison of RMSE and correlation coefficients for model error prediction using attributes and caption embeddings

| Dataset | RMSE w/ attributes | RMSE w/ caption embeddings | Spearman | Pearson |
|---|---|---|---|---|
| Waterbirds | 6.7311 | 6.6906 | 0.6159 | 0.5204 |
| CelebA | 2.0753 | 2.0677 | 0.7119 | 0.6497 |

### A.13.2 STATISTICAL SIGNIFICANCE

To statistically validate the performance of subsets in line with the hypotheses generated by the language model, we conduct t-tests across various hypotheses. We compare the observed accuracies of subsets where the hypothesized attributes are present to a null distribution designed to reflect a scenario where these attributes do not affect the classifier's accuracy.

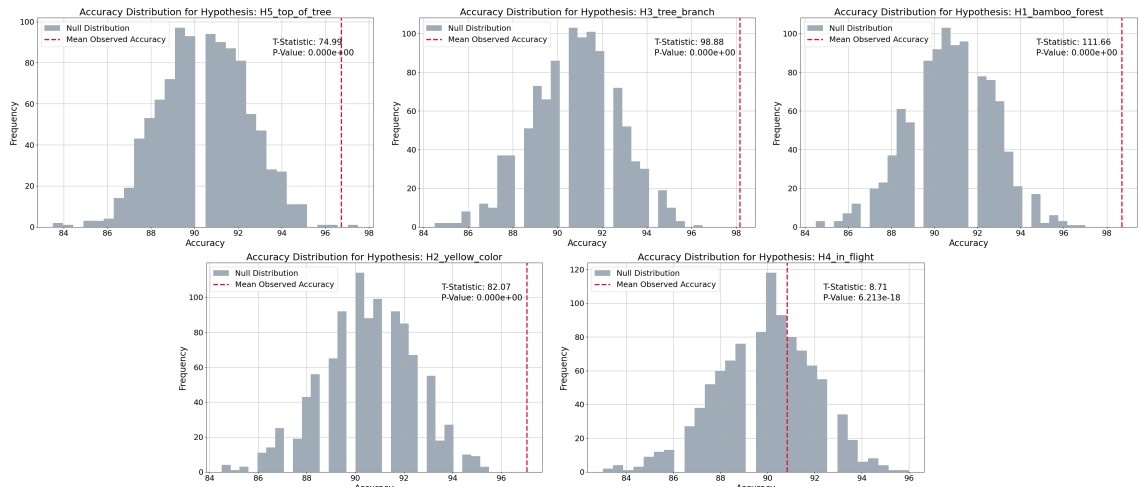

Figure 8: Statistical validation of hypotheses on the biases of RN Sup IN 1K-based classifier for *landbird* classification of Waterbirds dataset. Each panel represents a hypothesis tested, displaying the observed accuracy distribution against a null distribution. The t-statistics and p-values indicate significant differences, supporting the hypothesis that the presence of specific attributes significantly affects model accuracy.

For each hypothesis, we sample subsets from our dataset where the specific attribute mentioned in the hypothesis is present. This allows us to compute the mean observed accuracy for images exhibiting attributes relevant to each hypothesis. To construct a null distribution, we randomly sample an equal number of images, both with and without the attribute. This balanced sampling approach is crucial as it simulates the null hypothesis scenario that the attribute does not influence the classification accuracy.

The experiment is conducted with 1000 iterations for each hypothesis. In each iteration, 100 data points are randomly selected to calculate the classification accuracy against the ground truth. This repetitive sampling ensures robustness in our statistical testing by accurately approximating the distribution of accuracies under both observed and null conditions.

We apply a t-test (using `ttest_ind` from the `scipy.stats` package) for each hypothesis to compare the mean accuracies of the observed and null distributions. The t-test, chosen for its suitability in comparing means from two independent samples, provides a t-statistic and a p-value. These statistics quantify the evidence against the null hypothesis, offering a measure of the impact of the hypothesized attribute on model performance.

In our study, we utilize the RN Sup IN 1K model across various datasets, including Waterbirds, CelebA, NIH-CXR, and RSNA-Mammo, to evaluate the impact of identified attributes in each hypothesis on classification accuracy. Figures 9, 8, 10, 12, and 11 depict the t-test results for each dataset respectively.

Across all datasets, we observe consistently large t-statistics and extremely low p-values. This consistent pattern provides robust statistical evidence supporting the hypothesis that the attributes identified by LLM significantly influence the model's performance. The identified attributes, hypothesized by the LLM, correspond closely with the true underlying attributes that exhibit bias in the classifier $f$.

For each hypothesis, the observed mean accuracies in subsets of data where these attributes are present significantly exceeded those where these attributes were absent, as illustrated by the distinct separation in

the distributions shown in the figures. This validation confirms that these attributes are critical for achieving higher classification accuracy and that their proper identification and incorporation into model training can substantially reduce biases and improve the overall performance of the classifier.

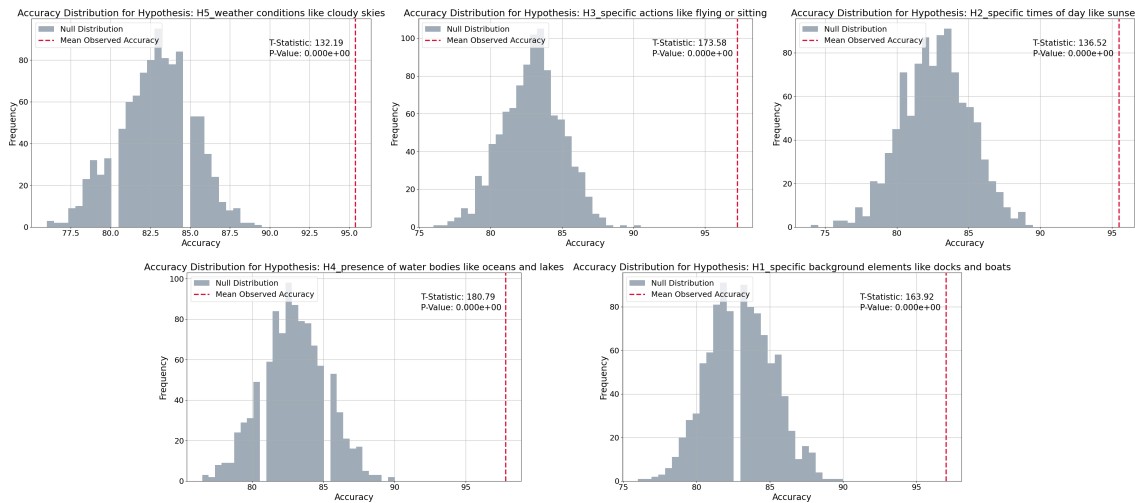

Figure 9: Statistical validation of hypotheses on the biases of RN Sup IN 1K-based classifier for *waterdbird* classification of Waterbirds dataset. Each panel represents a hypothesis tested, displaying the observed accuracy distribution against a null distribution. The t-statistics and p-values indicate significant differences, supporting the hypothesis that the presence of specific attributes significantly affects model accuracy.

### A.13.3 IMPACT OF DIFFERENT VISION LANGUAGE MODELS ON THE RETRIEVAL OF SENTENCES AND HYPOTHESIS GENERATION

Fig 13 compares the sentences retrieved in Sec. 2.1, signifying the attributes present in the correctly classified samples but missing in the misclassified samples. We perform this experiment using CXR-CLIP as VLR space pretrained with the MIMIC-CheXpert-ChestX-ray14 (MCC) and MIMIC-CheXpert (MC) datasets, respectively and retrieve top500 sentences. Also, Fig. 14 shows the hypothesis generated by LLM using the sentences retrieved using the two variants of CXR-CLIP. We observe that in both cases, the hypotheses identify chest tube or chest drain as a source of bias. Also, in both cases, the size and types of pneumothorax are highlighted. Notably, the MCC variant shows a substantially higher count of "chest tube" mentions, with 268 instances compared to the 114 mentions recorded by the MC variant (Fig. 15). This discrepancy highlights the influence of dataset composition and pretraining on our framework's ability to detect clinical attributes associated with pneumothorax in CXRs. The MCC variant integrates ChestX-ray14 data during the pertaining stage. So, it includes a broader variety of chest tube-related images. This experiment shows that the vision language model influences the formulation of the hypotheses.

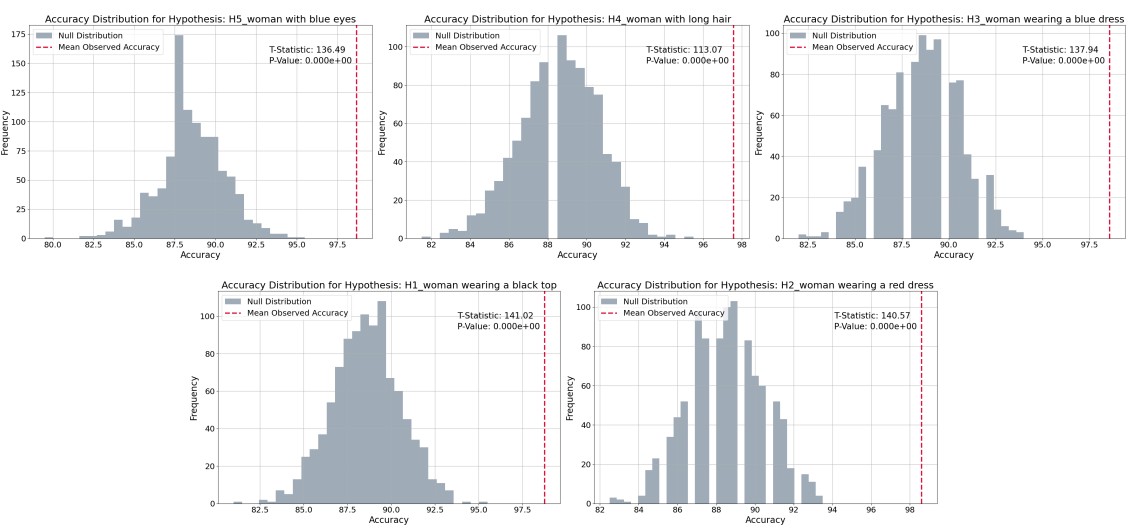

Figure 10: Statistical validation of hypotheses on the biases of RN Sup IN 1K-based classifier for *blond* classification of CelebA dataset. Each panel represents a hypothesis tested, displaying the observed accuracy distribution against a null distribution. The t-statistics and p-values indicate significant differences, supporting the hypothesis that the presence of specific attributes significantly affects model accuracy.

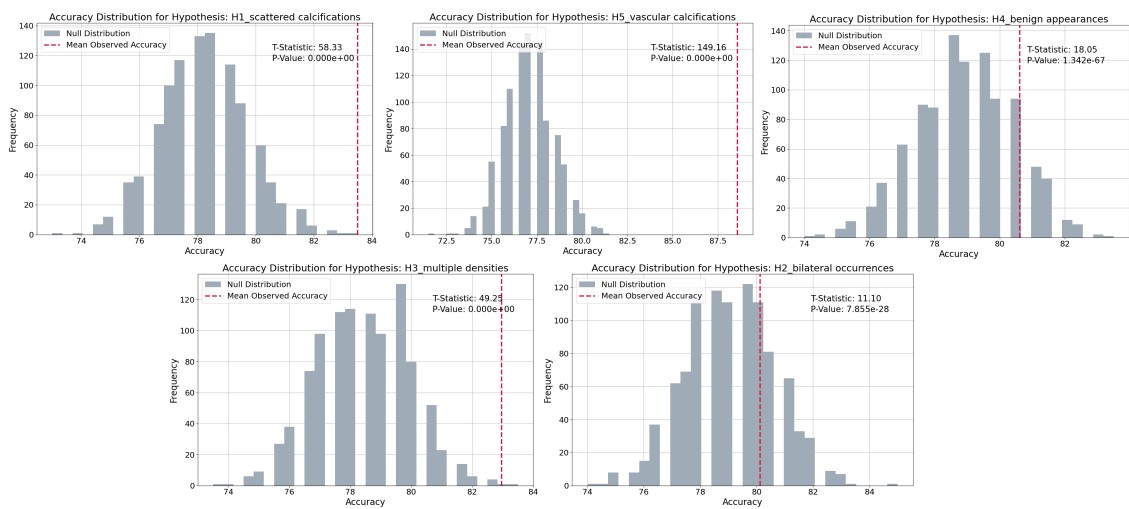

Figure 11: Statistical validation of hypotheses on the biases of classifier for *cancer* classification of the RSNA-Mammo dataset. Each panel represents a hypothesis tested, displaying the observed accuracy distribution against a null distribution. The t-statistics and p-values indicate significant differences, supporting the hypothesis that the presence of specific attributes significantly affects model accuracy.

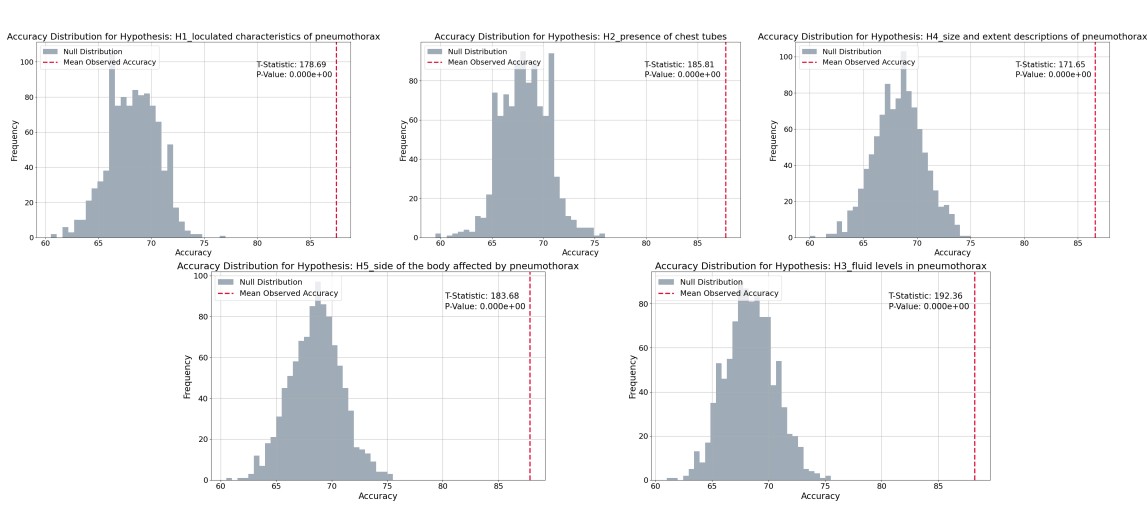

Figure 12: Statistical validation of hypotheses on the biases of RN Sup IN 1K-based classifier for *pneumothorax* classification of NIH-CXR dataset. Each panel represents a hypothesis tested, displaying the observed accuracy distribution against a null distribution. The t-statistics and p-values indicate significant differences, supporting the hypothesis that the presence of specific attributes significantly affects model accuracy.

**Sentences indicating the biased attributes using CXR-CLIP (M,C, C)**

1. interval placement of right apical and right base pleural drains with slight decrease in right hydropneumothorax
2. on ___ , patient had right thoracotomy and two apical and a basal pleural drains were placed and there was a substantial decrease in the volume of homogenous opacity in the right upper chest , presumably hematoma
3. two chest tubes remain in place in the right hemithorax with a persistent moderate to large right pneumothorax with apical pneumothorax component and basilar hydropneumothorax
4. two right chest tubes remain in place , with persistent moderate right pneumothorax , including apical pneumothorax component and basilar hydropneumothorax component
5. interval increase in size of the small right – sided pneumothorax with chest tube in place
6. moderate right pneumothorax despite the presence of three right chest tubes
7. in comparison with the study of ___ , there is been a a right middle lobectomy with 2 chest tubes in place and substantial pneumothorax
8. in comparison with study of ___ , there are now two chest tubes in place on the right with a small pneumothorax
9. right chest tubes remain in place , with persistent moderate – to – large right pneumothorax with apical pneumothorax component and basilar hydropneumothorax
10. extensive surgical changes are observed in the right lung with surgical sutures and three chest tubes , one apically and two basilary located
11. interval placement of a chest tube with a somewhat loculated pneumothorax in the right apex and the lateral lung in the area of recent surgery
12. increase in right apical pneumothorax with two chest tubes in place and no evidence of tension
13. in comparison with the earlier study of this date , there has been a small increase in the substantial right apical and basilar pneumothorax with the chest tube on water seal
14. in comparison with the study of ___ , there again is evidence of previous right upper lobectomy with two chest tubes in place and persistent pneumothorax
15. ap chest compared to ___ through ___ , 4 : 45 p . m .: moderate – to – large right pneumothorax improved between ___ and ___ and has been stable all day , despite presence of two right pleural tubes ending in the upper hemithorax
16. interval increase in size of a moderate to large right pneumothorax with the chest tubes on water seal
17. interval insertion of a right – sided chest tube in good position , right – sided hydro pneumothorax has changed with increased pleural air are and relative decrease of the pleural fluid
18. of the placing the left chest tube on water seal , there is a minimal increase in extent of the left postoperative predominantly basal pneumothorax

**Sentences indicating the biased attributes using CXR-CLIP (M,C)**

1. perhaps mild increase in hydropneumothorax but with chest tube remaining in place and no striking change
2. in comparison with the study of ___ , there is little change in the 3 left chest tubes with area of hydro pneumothorax persisting in the lateral aspect of the upper left chest as well as probably the left lung base
3. a moderate sized loculated hydropneumothorax demonstrates decrease in fluid component and increasing gas component , particularly in the right base
4. small right pleural effusion has replaced the previous basal pneumothorax that developed with previous drainage of pleural effusion and placement of 2 thoracostomy tubes
5. 2 right indwelling pleural drains are unchanged in their respective positions , and there has probably been some decrease in the volume of the right posterior air and pleural collection in the rib right lower hemi thorax
6. interval placement of right apical and right base pleural drains with slight decrease in right hydropneumothorax
7. other less likely possibility include expansion of known loculated hydropneumothorax ( chest tube does not appear to be draining this region )
8. increasing fluid within the multiple pockets of the pneumothorax on the right
9. decreased fluid and increased air in the right basilar hydropneumothorax , where the pleural catheter resides
10. moderate right pleural effusion with loculated hydro pneumothorax components is again demonstrated , with apparent slight increase in extent of right basilar hydro pneumothorax
11. three right – sided chest tubes remain in place with decrease in the loculated basilar hydropneumothoraces and some interval improvement in aeration at the right base
12. loculated right hydro pneumothorax , with right basilar pneumothorax component slightly increased
13. the only change is increasing fluid within a loculated hydropneumothorax , with corresponding decrease in air , located along the left lateral chest wall , of uncertain significance in the short – term postoperative course
14. multiple small loculated hydropneumothoraces are again demonstrated , with interval worsening of loculated hydropneumothoraces at the left base
15. fluid has now replaced air in the lateral component of the multi loculated left hydro pneumothorax , despite the insertion in that location of a new small drainage catheter
16. loculated he mall / hydro pneumothorax in the upper portion of the chest as well as in the left lung base are unchanged
17. small – to – moderate right hydropneumothorax , increase in the lateral costal fluid collection , and stable small apical air component
18. successful placement of chest tube and pleurx tube , small right basal loculated pneumothorax replaces area of successful pleural drainage

Figure 13: Comparing attribute identification using languages using different variants of CXR-CLIP Models. M, C, and C14 denote MIMIC-CXR, CheXpert, and ChestX-ray14, respectively. The left panel displays sentences retrieved from the MCC variant, while the right panel shows sentences from the MC variant. Both identify key attributes such as chest tubes and pneumothorax characteristics, with notable differences in the frequency of mentions, reflecting the impact of vision language pretrained models in identifying clinical features.

1551
1552
1553
1554
1555
1556
1557
1558
1559
1560
1561
1562
1563
1564
1565
1566
1567
1568
1569
1570
1571
1572
1573
1574
1575
1576
1577
1578
1579
1580
1581
1582

**Hypotheses with from sentences generate from CXR-CLIP (M,C, C)**

Hypotheses:
The classifier is making mistake as it is biased toward:
H1: chest tubes
H2: apical components
H3: descriptions of basilar components
H4: subcutaneous emphysema
H5: size descriptors of pneumothorax (small, moderate, large)

Prompt to test each hypotheses:
1. H1_chest_tubes: ['Chest x-ray showing pneumothorax with chest tubes in place.', 'Image of a lung with pneumothorax and visible chest tubes.', 'Chest radiograph of pneumothorax managed with chest tubes.', 'Diagnostic image displaying pneumothorax with chest tubes inserted.', 'Clinical chest x-ray with pneumothorax featuring chest tubes.']

2. H2_apical_components: ['Chest x-ray showing pneumothorax with apical component.', 'Image of a lung with pneumothorax affecting the apex.', 'Chest radiograph of pneumothorax with apical involvement.', 'Diagnostic image displaying pneumothorax focused on the lung apex.', 'Clinical chest x-ray with pneumothorax showing apical air collection.']

3. H3_basilar_components: ['Chest x-ray showing pneumothorax with basilar component.', 'Image of a lung with pneumothorax affecting the base.', 'Chest radiograph of pneumothorax with basal involvement.', 'Diagnostic image displaying pneumothorax focused on the lung base.', 'Clinical chest x-ray with pneumothorax showing basilar air collection.']

4. H4_subcutaneous_emphysema: ['Chest x-ray showing pneumothorax with subcutaneous emphysema.', 'Image of a lung with pneumothorax and visible subcutaneous air.', 'Chest radiograph of pneumothorax associated with subcutaneous emphysema.', 'Diagnostic image displaying pneumothorax with subcutaneous air pockets.', 'Clinical chest x-ray with pneumothorax and subcutaneous emphysema evident.']

5. H5_size_descriptors: ['Chest x-ray showing a small pneumothorax.', 'Image of a lung with a moderate pneumothorax.', 'Chest radiograph of a large pneumothorax.', 'Diagnostic image displaying a substantial pneumothorax.', 'Clinical chest x-ray with a minimal pneumothorax.']

**Hypotheses with from sentences generate from CXR-CLIP (M,C)**

Hypotheses:
The classifier is making mistake as it is biased toward:
H1: loculated characteristics of pneumothorax
H2: presence of chest tubes
H3: fluid levels in pneumothorax
H4: size and extent descriptions of pneumothorax
H5: side of the body affected by pneumothorax

Prompt to test each hypotheses:
1. H1_loculated characteristics of pneumothorax:
['Chest X-ray showing loculated pneumothorax with varying air and fluid levels', 'Loculated air pockets in pneumothorax as seen in a chest radiograph', 'Pneumothorax with loculated air collections complicating the diagnosis', 'Loculated pneumothorax with complex air and fluid separation', 'Detailed view of loculated pneumothorax with chest tube intervention']

2. H2_presence of chest tubes:
['Chest X-ray with visible chest tubes in place for pneumothorax treatment', 'Pneumothorax management with chest tubes as seen in the radiograph', 'Chest radiograph depicting the placement of chest tubes in pneumothorax', 'Influence of chest tubes on the appearance of pneumothorax in X-ray images', 'Chest tubes in situ for a patient with pneumothorax on the radiograph']

3. H3_fluid levels in pneumothorax:
['Chest X-ray showing pneumothorax with significant fluid levels', 'Pneumothorax with varying degrees of fluid accumulation in chest X-ray', 'Radiographic appearance of pneumothorax with fluid levels', 'Assessment of fluid levels in pneumothorax via chest radiography', 'Fluid levels indicating severity of pneumothorax in a chest X-ray']

4. H4_size and extent descriptions of pneumothorax:
['Chest X-ray showing a large pneumothorax covering extensive lung area', 'Moderate sized pneumothorax visible on the right side in chest X-ray', 'Small apical pneumothorax detected in a routine chest X-ray', 'Extent of pneumothorax as a critical factor in chest X-ray analysis', 'Evaluating the size and spread of pneumothorax in chest radiographs']

5. H5_side of the body affected by pneumothorax:
['Right-sided pneumothorax as shown in chest X-ray imaging', 'Left basal pneumothorax detected in a diagnostic chest X-ray', 'Chest X-ray revealing pneumothorax on the left side of the chest', 'Comparison of right and left side pneumothorax in X-ray images', 'Implications of pneumothorax location on the left side in chest X-rays']

Figure 14: Comparison of hypothesis generation and testing using different variants of CXR-CLIP. We highlight the common attributes for both variants in yellow.

1583
1584
1585
1586
1587
1588
1589
1590
1591
1592
1593
1594
1595
1596
1597

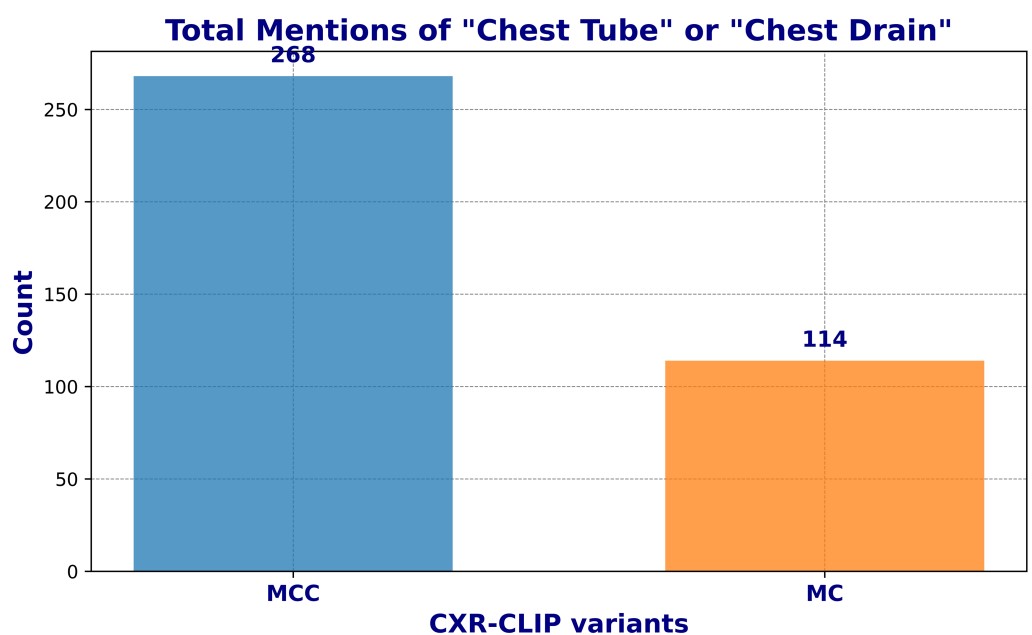

Figure 15: Comparison of "chest tube" and "chest drain" mentions in sentences retrieved using different CXR-CLIP variants. The MCC variant shows a significantly higher count, highlighting differences in dataset composition during the pretraining and its impact on the model's bias detection.

### A.13.4 CLOSEST HYPOTHESIS TO THE GROUND TRUTH ATTRIBUTE

Tables 8 and 7 show the `top3` hypotheses for RN Sup IN1K (convolution-based) and ViT Sup IN1K (transformer-based) architectures, respectively. These hypotheses are the most similar to the ground truth attribute on which the source model $f$ is biased.

Table 7: Top 3 associated hypotheses for the ground truth biased attribute for ViT Sup IN1K model on various datasets

| Dataset (Label) | Attribute | Top 3 hypotheses |
|---|---|---|
| Waterbirds (waterbird) | Water | 1. activities like swimming or flying
2. conditions like cloudy or sunny
3. presence of objects like boats or rocks |
| Waterbirds (landbird) | Land | 1. bird in the middle of a forest
2. yellow bird
3. bird sitting on top of a tree |
| CelebA (Blonde) | Women | 1. woman wearing red dress
2. woman with red top
3. black jacket |
| MetaShift (Dog) | Outdoor | 1. presence of a leash
2. presence of a ball
3. presence of a car |
| MetaShift (Cat) | Indoor | 1. beds
2. windows
3. televisions |

### A.13.5 RESULTS ON ACCGAP

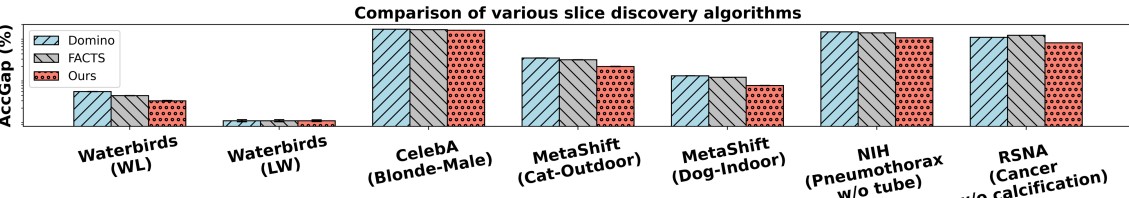

Figure 16: Comparisons of `AccGap` across various datasets for CNN-based slice discovery algorithms. Lower `AccGap` values indicate better performance in approximating true error slices, plotted on a logarithmic scale for clarity. WL and LW denote waterbirds on land and vice versa, respectively.

Figure 16 illustrates the `AccGap` for all slice discovery methods across CNN models. `AccGap` measures the accuracy gap between the ground truth and predicted error slices. For *e.g.,* in MetaShift, images of `dogs` are biased by the `outdoor` attribute. For the RN Sup IN1k model, hypotheses generated by the LLM (Sec. 2.2) identify `televisions`, `windows`, and `beds` as the biased attributes most associated with

Table 8: Top 3 associated hypotheses for the ground truth biased attribute for RN Sup IN1K model on various datasets

| Dataset (Label) | Attribute | Top 3 hypotheses |
|---|---|---|
| Waterbirds (waterbird) | Water | 1. water bodies like oceans and lakes
2. actions like flying or sitting
3. conditions, e.g., cloudy skies |
| Waterbirds (landbird) | Land | 1. bird being in flight
2. bird perching on top of a tree
3. bird perching on a tree branch |
| CelebA (Blonde) | Women | 1. woman with long hair
2. woman wearing red dress
3. a black jacket |
| MetaShift (Dog) | Outdoor | 1. dogs in motion
2. dogs on leashes
3. beach environments |
| MetaShift (Cat) | Indoor | 1. televisions
2. windows
3. beds |
| NIH (pneumothorax) | Chest tube | 1. the presence of chest tubes
2. loculated pneumothorax
3. size and extent of pneumothorax |
| RSNA-Mammo (cancer) | Calcification | 1. scattered calcifications
2. vascular calcifications
3. bilateral occurrences |

`outdoor` (in Tab. 8). We compute the `AccGap` with images of the predicted error slice (*i.e.,* images of `dogs` lacking the predicted attributes) and the ground truth slice (*i.e.,* images without `outdoor` attributes). LADDER consistently exhibits the lowest `AccGap`, indicating that the model $f$ underperforms similarly on the discovered error slices as on the ground truth slices.

### A.13.6    Extended qualitative results for our slice discovery method on various datasets

Figures 17 and 18 report LLM-generated the list of hypotheses and the prompts to test them discussed in the Sec. 4. Figures 19, 20, 21, 22, and 23 illustrate qualitative results of our method applied on various datasets using RN Sup IN1 models. Specifically, they showcase the classification of pneumothorax patients from NIH, "landbird" from the Waterbirds, "blond" from CelebA, "cat" and "dog" from MetaShift, and "cancer" from the RSNA-Mammo datasets, respectively. Also, Figures 25, 24, 26, and 27 depict similar qualitative results for "landbird" and "waterbird" from the Waterbirds dataset, as well as "blond" from CelebA for ViT Sup IN1k classifiers. In all the cases, LADDER correctly identifies the hypothesis with true attribute causing biases in the given classifier $f$.

```
1. H1_loculated characteristics of pneumothorax:
 ['Chest X-ray showing loculated pneumothorax with varying air and fluid levels',
'Loculated air pockets in pneumothorax as seen in a chest radiograph', 'Pneumothorax
with loculated air collections complicating the diagnosis', 'Loculated pneumothorax
with complex air and fluid separation', 'Detailed view of loculated pneumothorax with
chest tube intervention']

2. H2_presence of chest tubes:
['Chest X-ray with visible chest tubes in place for pneumothorax treatment',
'Pneumothorax management with chest tubes as seen in the radiograph', 'Chest
radiograph depicting the placement of chest tubes in pneumothorax', 'Influence of
chest tubes on the appearance of pneumothorax in X-ray images', 'Chest tubes in situ
for a patient with pneumothorax on the radiograph']

3. H3_fluid levels in pneumothorax:
['Chest X-ray showing pneumothorax with significant fluid levels', 'Pneumothorax with
varying degrees of fluid accumulation in chest X-ray', 'Radiographic appearance of
pneumothorax with fluid levels', 'Assessment of fluid levels in pneumothorax via
chest radiography', 'Fluid levels indicating severity of pneumothorax in a chest X-
ray']

4. H4_size and extent descriptions of pneumothorax:
['Chest X-ray showing a large pneumothorax covering extensive lung area', 'Moderate
sized pneumothorax visible on the right side in chest X-ray', 'Small apical
pneumothorax detected in a routine chest X-ray', 'Extent of pneumothorax as a
critical factor in chest X-ray analysis', 'Evaluating the size and spread of
pneumothorax in chest radiographs']

5. H5_side of the body affected by pneumothorax:
['Right-sided pneumothorax as shown in chest X-ray imaging', 'Left basal pneumothorax
detected in a diagnostic chest X-ray', 'Chest X-ray revealing pneumothorax on the
left side of the chest', 'Comparison of right and left side pneumothorax in X-ray
images', 'Implications of pneumothorax location on the left side in chest X-rays']
```

Figure 17: Hypothesis and prompts by LADDER RN Sup IN1k-based classifier for *pneumothorax* classification in **NIH** dataset. LADDER uses these prompts to test the hypothesis. We highlight the hypothesis generated by LADDER that corresponds to the ground truth biased attribute (*e.g.,* chest tubes for pneumothorax) in **yellow**.

### A.13.7 EXTENDED RESULTS ON COMPARING DIFFERENT ERROR MITIGATION STRATEGIES USING VIT SUP IN1K-BASED MODELS

Tab. 9 compares different error mitigation algorithms for ViT Sup IN1k based models ($f$).

### A.13.8 IMPROVEMENT ON THE ZERO-SHOT ACCURACY OF VISION LANGUAGE MODELS USING THE ATTRIBUTES FROM THE EXTRACTED HYPOTHESIS BY LADDER

To evaluate the impact of LADDER's attribute-based slice discovery on zero-shot performance, we conducted experiments using a CLIP-based vision-language model across multiple datasets. LADDER extracts fine-

Table 9: Benchmarking error mitigation methods over 3 seeds for ViT models pretrained with IN1k using the supervised method (RN Sup IN1k). We bold-face and underline the best and second-best results, respectively.

| Method | Waterbirds | | CelebA | |
| --- | --- | --- | --- | --- |
| | Mean(%) | WGA(%) | Mean(%) | WGA(%) |
| Vanilla (ERM) | 82.7 | 51.2 | 95.2 | 46.8 |
| Mixup | 81.8 | 44.9 | **95.8** | 48.3 |
| IRM | 79.8 | 54.5 | 85.1 | 48.7 |
| MMD | 83.6 | 42.5 | 95.6 | 54.2 |
| JTT | 81.7 | 49.1 | 94.8 | 52.7 |
| GroupDRO | 82.2 | 53.1 | 93.5 | 80.1 |
| CVaRDRO | 83.5 | 46.6 | 95.6 | 55.1 |
| LISA | 83.7 | 48.8 | 95.6 | 60.2 |
| $DFR_{val}$ | 85.0 | 76.2 | 91.3 | 81.1 |
| $Ladder_{IPTW}$ (**ours**) | 85.1 | 81.4 | 89.1 | **83.8** |
| $Ladder_{bal}$ (**ours**) | **85.3** | **86.5** | 90.7 | 83.4 |

grained attributes from error-prone data slices, which we incorporated as detailed prompts for zero-shot classification. These prompts were generated from hypotheses produced by the LADDER framework and reflect nuanced characteristics of the data that a model might otherwise overlook. We compare these attribute-driven prompts against standard, baseline prompts typically used for zero-shot tasks.

**Experimental Process.** For each dataset, we implemented two types of zero-shot prompts:

- **Baseline prompts**: CLIP-based prompts (Radford et al., 2021) *e.g.,* [`a photo of a landbird` and `a photo of a waterbird`] for the Waterbirds dataset for natural images, CXR-CLIP (You et al., 2023) prompts *e.g.,* [`no pneumothorax`, `pneumothorax`] for NIH, Mammo-CLIP (Ghosh et al., 2024) prompts *e.g.,* [{`no cancer, no malignancy`}, {`cancer, malignancy`}] for RSNA-Mammo and VinDr-Mammo.

- **LADDER-derived prompts**: These prompts were generated based on the attributes extracted from LADDER's hypotheses, providing a more detailed description of the data. For example, in the Waterbirds dataset, we used prompts like `a photo of a waterbird on docks and boats` or `a photo of a landbird inside on bamboo forest`. In this experiment, we use the attributes from the hypotheses extracted from RN Sup IN1k (Resnet 50 pretrained with ImageNet 1K and supervised learning) classifier.

We evaluated the zero-shot classification performance of the model using both prompt types. The results are shown in Tab. 10.

**Results.** The results demonstrate a significant improvement in zero-shot accuracy when using LADDER-extracted attributes as prompts. Across all datasets, the attribute-driven prompts outperformed the baseline, indicating the effectiveness of using detailed, hypothesis-driven attributes to enhance zero-shot performance. In the **Waterbirds** dataset, LADDER prompts improved accuracy by +8.56%, rising from 50.40% with baseline prompts to 58.96% with LADDER attributes. The improvement was even more pronounced for the **NIH** dataset, with a +19.05% gain (49.17% to 68.22%). The **RSNA** dataset also saw a notable improvement, with a +5.81% gain in accuracy (60.17% to 65.98%). The improvements for **CelebA** (+0.32%) and **VinDr** (+1.41%) were more modest but still indicate that using LADDER's attribute-based prompts provides consistent gains across various domains. These results highlight the ability of LADDER to extract meaningful

attributes that guide the vision-language model to more accurate predictions, even in zero-shot settings where explicit training on the target data is absent. By leveraging these hypotheses, LADDER enables more precise alignment between image representations and class descriptions, significantly enhancing zero-shot performance.

Table 10: Application1: Boost in Zero-shot accuracy results using attributes from the hypotheses extracted from RN Sup IN1k (Resnet 50 pretrained with ImageNet 1K and supervised learning) classifier

| Dataset | CLIP Prompts | LADDER Hypotheses | Gain |
|---------|--------------|-------------------|------|
| Waterbirds | 50.40 | **58.96** | +8.56 ↑ |
| CelebA | 86.69 | **87.01** | +0.32 ↑ |
| NIH | 49.17 | **68.22** | +19.05 ↑ |
| RSNA | 60.17 | **65.98** | +5.81 ↑ |
| VinDr | 90.92 | **92.33** | +1.41 ↑ |

### A.13.9 CLIP SCORE COMPARISON OF VARIOUS ATTRIBUTES EXTRACTED BY LADDER

Refer to Fig. 28 for the CLIP scores (discussed in Appendix A.5) of various attributes extracted from the hypotheses by LADDER. For *e.g.,* the correctly classified samples for the waterbird class in the Waterbirds dataset have a bias on the water-related backgrounds. As a result, the CLIP score of `ocean, boat, lake` is high. We observe consistent results for other datasets as well.

### A.13.10 IMPROVEMENT ON DIFFERENT SLICES OF URBANCARS BENCHMARK

Tab. 11 shows that LADDER achieves higher accuracy compared to the Whac-A-Mole method(Li et al., 2023b) across multiple shortcut benchmarks on the Urbancars dataset, without prior knowledge of the number or types of possible shortcuts.

Table 11: LADDER achieves higher accuracy compared to the Whac-A-Mole method(Li et al., 2023b) across multiple shortcut benchmarks on the Urbancars dataset, without prior knowledge of the number or types of possible shortcuts.

| Method | Mean Acc | BG gap | CoObj Gap | BG+CoObj Gap |
|--------|----------|--------|-----------|--------------|
| ERM | 96.4 | -15.3 | -11.2 | -69.2 |
| Whac-A-Mole | 95.2 | -2.4 | -2.9 | -5.8 |
| Whac-A-Mole + LADDER Hypothesis | 95.6 | -1.8 | -2.8 | -4.6 |
| LADDER $_{IPTW}$ | 92.2 | -1.1 | -1.6 | -3.8 |

### A.13.11 EXTENDED RESULTS ON DISCOVERED HYPOTHESIS BY LADDER FOR VARIOUS ARCHITECTURES AND PRE-TRAINING METHODS

Fig. 29 illustrates additional results for the CelebA and Metashift datasets, demonstrating that LADDER accurately captures various sources of bias, regardless of the underlying architectures or pre-training methods.

### A.13.12 EXTENDED RESULTS ON BREAST DATASETS FOR WGA USING SLICES BY DOMINO, FACTS AND LADDER

Fig. 30 shows LADDER improves WGA compared to other bias mitigation methods for RSNA-Mammo and VinDr-Mammo datasets.

### A.13.13 ABLATION 1: WGA OF LADDER USING OTHER CAPTIONING METHODS

Tab. 12 presents an ablation study evaluating the effect of various captioning models on LADDER's performance in mitigating biases. The quality of captions directly affects LADDER's ability to effectively generate hypotheses, as these captions are analyzed by LLMs to identify biased attributes contributing to model errors. LADDER then pseudo-labels these attributes to systematically mitigate the identified biases. We consider different captioning models, including BLIP (Li et al., 2022), BLIP2 (Li et al., 2023a), ClipCap (Mokady et al., 2021), and GPT-4o (Wu et al., 2024), with **ResNet Sup IN1k** as the classifier.

The results indicate that the more advanced captioning model, GPT-4o, significantly improves LADDER's performance, achieving the highest Worst Group Accuracy (WGA) and mean accuracy across both datasets. Specifically, GPT-4o achieves a WGA of 94.5% on Waterbirds and 91.9% on CelebA, which is substantially better than the other models. BLIP and BLIP2 demonstrate comparable results, with BLIP slightly outperforming BLIP2 in the Waterbirds dataset, while BLIP2 performs better on CelebA in WGA. In contrast, ClipCap consistently yields the lowest scores, implying that simpler captioning methods are less effective for enhancing LADDER's bias identification capabilities. Overall, the results underscore the importance of selecting a high-quality captioning model to maximize LADDER's effectiveness. While more sophisticated models like GPT-4o entail higher costs, their significant impact on bias mitigation performance, particularly on WGA, makes them an indispensable choice in scenarios where accuracy is critical.

Table 12: Ablation 1: Ablation study with different captioning models and its impact on LADDER's performance. We use RN Sup IN1k as the classifier for both the datasets.

| | Waterbirds | | CelebA | |
|---|---|---|---|---|
| Method | Mean Acc | WGA | Mean Acc | WGA |
| BLIP (Li et al., 2022) | 92.1 | 93.7 | 89.7 | 90.2 |
| BLIP2 (Li et al., 2023a) | 92.3 | 93.2 | 89.1 | 90.5 |
| ClipCap (Mokady et al., 2021) | 91.7 | 92.7 | 87.3 | 88.4 |
| GPT-4o (Wu et al., 2024) | **92.8** | **94.5** | **90.6** | **91.9** |

### A.13.14 ABLATION 2: SLICE DISCOVERY BY LADDER USING DIFFERENT LLMS

In this ablation study, we explore how different LLMs impact the effectiveness of LADDER in discovering data slices and generating hypotheses for bias identification. We aim to discover the biases from RN Sup IN1k classifier for natural images and CXRs, and EN-B5 classifier for mammograms. We utilize four LLMs: GPT-4o, Claude 3.5 Sonnet, LLaMA 3.1 70B, and Gemini 1.5 Pro. Fig. 31 illustrates the different attributes these models highlight across multiple datasets, including Waterbirds, CelebA, NIH, RSNA, VinDr, and MetaShift. Each LLM aims to extract a hypothesis related to an attribute, signifying the classifier's mistake. These attributes potentially lead to systematic model biases. As shown in Fig. 31, each LLM focuses on distinct subsets of attributes, reflecting their unique interpretation capabilities. Despite these differences, there is significant overlap in the overall hypotheses generated across the models, indicating consistency in identifying the attributes contributing to model errors.

For instance, in the Waterbirds dataset, all LLMs frequently highlight attributes like `ocean` and `boat` for the waterbird class and `bamboo forest` and `tree branch` for the landbird class. These attributes align closely with the ground truth bias in this dataset, which relates to water and land backgrounds being associated with the respective bird classes. This suggests that LLMs effectively identify these underlying environmental biases that lead to systematic errors. Similarly, in medical datasets, such as NIH-CXR for pneumothorax, all LLMs consistently highlight `chest tube` as a common attribute for misclassified samples. This reflects a true bias, as the presence of a chest tube often strongly correlates with pneumothorax cases. Identifying this attribute helps understand the systematic bias that models may develop when chest tubes are spuriously correlated in pneumothorax images.

This consistency across various LLMs demonstrates the robustness of LADDER for systematic bias detection, irrespective of the underlying LLM used. The results highlight that LADDER is effective at leveraging the strengths of different LLMs to produce meaningful insights into model behavior, regardless of which LLM is utilized. Moreover, it emphasizes the versatility of using LLMs for extracting domain-specific attributes—whether the focus is on natural images, chest X-rays, or mammography scans—while maintaining cost efficiency and avoiding manual annotation. Overall, this ablation shows that the specific choice of LLM slightly influences which attributes are emphasized, but all models effectively support the generation of comprehensive hypotheses that capture the biases inherent in different datasets.

### A.13.15   ABLATION 3: WGA BY LADDER USING THE HYPOTHESIS BY DIFFERENT LLMs

Fig. 32 illustrates the worst group accuracy (WGA) achieved across multiple datasets when utilizing LADDER to mitigate biases with different LLMs. The LLMs compared in this study include Claude 3.5 Sonnet, LLaMA 3.1 70B, Gemini 1.5 Pro, and GPT-4o. We consider the RN Sup IN1k classifier for natural images and CXRs, as well as the EN-B5 classifier for mammograms. The primary aim of this ablation is to assess how well LADDER can mitigate biases when generating hypotheses using different LLMs. As shown in Fig. 32, the WGA values remain consistently high across all LLMs, indicating that LADDER is effective in mitigating biases irrespective of the choice of LLM for hypothesis generation. Specifically, all LLMs achieve WGA scores of over 80% for most datasets, with only slight variations between models. This consistency demonstrates the robustness of LADDER in leveraging different LLMs to address model biases effectively. For datasets like Waterbirds and CelebA, the performance across all LLMs is nearly identical, suggesting that the generated hypotheses successfully capture the underlying biases and lead to similar improvements in fairness. In medical datasets, such as NIH and RSNA, the trend is also maintained, with LLMs like GPT-4o and Gemini 1.5 Pro achieving better results than other LLMs. These findings emphasize that the specific choice of LLM has only a minor impact on the overall ability of LADDER to mitigate bias. This makes LADDER a flexible and cost-effective solution, as it can work effectively with a range of LLMs, each with different computational costs and capabilities. Using different LLMs ensures flexibility based on resource availability while effectively identifying and mitigating dataset biases.

**1. H1_specific background elements like docks and boats:** ['a seagul sitting on a dock with boats in the background', 'a bird sitting on the edge of a dock at night with boats nearby', 'a seagul flying over the water with a boat in the background', 'a bird perched on top of a boat in the ocean', 'a seagul on the beach with boats in the background']

**2. H2_specific times of day like sunset:** ['a seagull sitting on a rock in front of a lighthouse at sunset', 'a seagul flying over the ocean at sunset', 'a yellow flower floating in the ocean at sunset', 'a bird flying over the ocean with a sunset in the background', 'a bird perched on a rock in the ocean at sunset']

**3. H3_specific actions like flying or sitting:** ['a seagul catching a fish in the ocean while flying', 'a bird flying over the ocean', 'a seagul sitting on a rock in the ocean', 'a bird sitting on top of an iceberg', 'a seagul sitting on a wooden post in front of a body of water']

**4. H4_presence of water bodies like oceans and lakes:** ['a duck swimming in the ocean', 'a seagul in the water with its wings spread out', 'a bird standing on the beach with the ocean in the background', 'a bird flying over the ocean with waves in the background', 'two seaguls sitting on rocks by the water in black and white']

**5. H5_weather conditions like cloudy skies:** ['a rock in the water with a cloudy sky in the background', 'a bird flying over the ocean on a cloudy day', 'a duck on the beach with a dark sky in the background', 'a bird flying over the water on a beach with cloudy skies', 'a bird sitting on the beach with cloudy skies in the background']

Figure 18: Hypothesis and prompts by LADDER RN Sup IN1k-based classifier for *waterbird* classification in **Waterbirds** dataset. LADDER uses these prompts to test the hypothesis. We highlight the hypothesis generated by LADDER that corresponds to the ground truth biased attribute (*e.g.,* water for waterbirds) in **yellow**.

**❶** Sentences indicating the biased attributes

```
1. a yellow bird perched on a branch in a bamboo forest
2. a bird perched on a tree branch in a bamboo forest
3. a cardinal bird in a bamboo forest
4. a bird perched on a tree in a bamboo forest
5. a bird perched on a branch in a bamboo forest
6. a bird perched on top of a tree in a bamboo forest
7. a bird sitting on a branch in a bamboo forest
8. a bird sitting on top of a tree in a bamboo forest
9. a yellow bird perched on a tree branch in the woods
10. a yellow bird in the bamboo forest
11. a bird perched on a bamboo tree in a bamboo forest
12. a yellow bird perched on a tree in the rainforest
….
```

**❸** Performance of the model on slices where the biased attribute is present/absent

**H1: Specific setting of a bamboo forest:**

Present: 98.0 %
Absent: 81.6 %

**H2: Presence of yellow color:**

Present: 97.0 %
Absent: 83.9 %

**H3: The bird perching on a tree branch:**

Present: 98.2 %
Absent: 83.0 %

**H4: The bird being in flight:**

Present: 90.7 %
Absent: 89.1 %

**H5: The bird perching on top of a tree:**

Present: 96.6 %
Absent: 84.2 %

**❷** Hypotheses with biased attributes generated by LLM

```
Hypotheses:
The classifier is making mistake as it is biased toward:
H1: specific setting of a bamboo forest
H2: presence of yellow color
H3: the bird perching on a tree branch
H4: the bird being in flight
H5: the bird perching on top of a tree

Prompt to test each hypotheses:
1. H1_specific setting of a bamboo forest:  ['A bird perched in a
dense bamboo forest', 'A bird flying through a bamboo forest', 'A
bird sitting on a bamboo tree in a lush green bamboo forest', 'A
bird standing on the ground surrounded by bamboo stalks', 'A bird
in the middle of a bamboo forest looking around']

2. H2_presence of yellow color: ['A yellow bird perched quietly
on a branch', 'A bright yellow bird flying against a clear sky',
'A small yellow bird sitting on a tree branch', 'A yellow bird
standing on the ground among fallen leaves', 'A vivid yellow bird
in the middle of a green forest']

3. H3_the bird perching on a tree branch: ['A bird perched on a
thin tree branch', 'A bird resting on a branch of an oak tree',
'A small bird holding onto a swaying branch', 'A bird perched on
a leafless tree branch in winter', 'A bird sitting quietly on a
branch in a serene forest']

4. H4_the bird being in flight: ['A bird soaring high in the
sky', 'A bird in mid-flight over the treetops', 'A bird flying
low over a river', 'A bird gliding in the air with wings spread
wide', 'A bird in flight chasing after insects']

5. H5_the bird perching on top of a tree: ['A bird perched at the
very top of a tall tree', 'A bird surveying its surroundings from
the top of a tree', 'A bird on top of a pine tree singing', 'A
bird at the highest branch of a tree looking down', 'A bird
resting at the peak of a tree during sunset']
```

Figure 19: LADDER discovers slices for biased attributes in RN Sup IN1k-based classifier for *landbird* classification in **Waterbirds** dataset. This figure details the slice discovery process for biased attributes involving sentence analysis, hypothesis generation by an LLM, and the model's performance on slices where attributes are present or absent, demonstrating how biases affect classifier accuracy. We highlight the hypothesis generated by LADDER that corresponds to the ground truth biased attribute (*e.g.,* land for landbirds) in **yellow**.

❶ Sentences indicating the biased attributes

1. a woman with **long blonde** hair **smiling** and wearing a **black top**
2. a woman with **long blonde** hair standing in front of a **red carpet**
3. a woman with **long blonde** hair standing on a **red carpet**
4. a girl with **long blonde** hair in a black frame on a **white background**
5. a woman with **long blonde** hair standing in front of a **stone wall**
6. a woman with **long blonde** hair is smiling and holding a pillow
7. a woman with **long blonde** hair posing in front of a christmas tree
8. a woman with **long blonde** hair standing in front of a christmas tree
9. a woman with blonde hair standing on **a red carpet**
10. a woman with **long blonde** hair and **blue eyes smiles** at the camera
11. a woman with **long blonde** hair and **a black jacket smiles** at the camera
12. a woman with **long blonde** hair in front of a **white** background
....

❸ Performance of the model on slices where the biased attribute is present/absent

H1: black top:

Present: 99.4 %
Absent: 79.5 %

H2: Woman wearing a red dress:

Present: 98.0 %
Absent: 80.1 %

H3: blue dress:

Present: 98.6 %
Absent: 80.4 %

H4: Woman with long hair:

Present: 97.8 %
Absent: 79.5 %

H5: blue eyes:

Present: 99.2 %
Absent: 78.7 %

❷ Hypotheses with biased attributes generated by LLM

```
Hypotheses:
The classifier is making mistake as it is biased toward:
H1: black top
H2: woman wearing a red dress
H3: a blue dress
H4: woman with long hair
H5: blue eyes

Prompt to test each hypotheses:
1. H1_black top:
['A woman with blonde hair wearing a black top', 'Blonde woman in
a black top', 'A blonde female wearing a black top smiling',
'Portrait of a blonde woman wearing a black top', 'A young woman
with blonde hair dressed in a black top']

2. H2_woman wearing a red dress:
['A woman with blonde hair wearing a red dress', 'Blonde woman in
a red dress', 'A blonde female wearing a red dress smiling',
'Portrait of a blonde woman wearing a red dress', 'A young woman
with blonde hair dressed in a red dress']

3. H3_blue dress:
['A woman with blonde hair wearing a blue dress', 'Blonde woman
in a blue dress', 'A blonde female wearing a blue dress smiling',
'Portrait of a blonde woman wearing a blue dress', 'A young woman
with blonde hair dressed in a blue dress']

4. H4_woman with long hair:
['A woman with blonde long hair', 'Blonde woman with long flowing
hair', 'A blonde female with long hair', 'Portrait of a blonde
woman with long hair', 'A young woman with long blonde hair']

5. H5_blue eyes:
['A woman with blonde hair and blue eyes', 'Blonde woman with
blue eyes', 'A blonde female with blue eyes looking at the
camera', 'Portrait of a blonde woman with blue eyes', 'A young
woman with blonde hair and blue eyes']
```

Figure 20: LADDER discovers slices for biased attributes in RN Sup IN1k-based classifier for *blond* classification in **CelebA** dataset. This figure details the slice discovery process for biased attributes involving sentence analysis, hypothesis generation by an LLM, and the model's performance on slices where attributes are present or absent, demonstrating how biases affect classifier accuracy. We highlight the hypothesis generated by LADDER that corresponds to the ground truth biased attribute (*e.g.,* woman for blond) in **yellow**.

**❶** Sentences indicating the biased attributes

1. a cat **sleeping** on top of a **computer mouse**
2. a cat **sleeping** on a **desk** in an **office**
3. a cat **sleeping** on top of a **laptop computer**
4. a cat **laying** on top of a **computer keyboard**
5. a cat **laying** on top of a **computer** on a **desk**
6. a cat **laying** on top of a **laptop computer**
7. a cat **sleeping** on top of a **laptop**
8. a cat **laying** in a **bathroom sink**
9. a cat **laying** on a **laptop computer**
10. a cat **laying** on top of a laptop on a **bed**
11. a cat **sitting** in a sink in a **bathroom**
12. a cat **laying** on a bed next to a **book**

....

**❸** Performance of the model on slices where
the biased attribute is present/absent

**H1: Laptops:**

Present: 99.4 %
Absent: 83.5 %

**H2: Bathroom settings:**

Present: 99.0 %
Absent: 83.1 %

**H3: Beds:**

Present: 98.3 %
Absent: 84.4 %

**H4: Desks:**

Present: 99.4 %
Absent: 82.5 %

**H5: Sinks:**

Present: 99.0 %
Absent: 81.7 %

**❷** Hypotheses with biased attributes generated by LLM

```
Hypotheses:
The classifier is making mistake as it is biased toward:
H1: laptops
H2: bathroom settings
H3: beds
H4: desks
H5: sinks
H6: sitting positions

Prompt to test each hypotheses:
1. H1_laptops: ['a cat sitting on a laptop in a bright room', 'a
cat lying on a laptop on a desk', 'a cat sleeping next to a
laptop on a couch', 'a cat playing with a laptop in a home
setting', 'a cat watching the screen of a laptop']

2. H2_bathroom settings: ['a cat sitting on a bathroom counter',
'a cat playing with a shower curtain', 'a cat sleeping on a
bathroom rug', 'a cat exploring a bathroom shelf', 'a cat inside
a bathtub']

3. H3_beds: ['a cat lying on a bed with pillows', 'a cat playing
on a bed next to a window', 'a cat sleeping on a large bed in a
bedroom', 'a cat sitting on a bed next to a book', 'a cat under
the blanket on a bed']

4. H4_desks: ['a cat sitting on a wooden desk', 'a cat lying
under a desk lamp', 'a cat playing with papers on a desk', 'a cat
napping on a cluttered desk', 'a cat watching a mouse on a desk']

5. H5_sinks: ['a cat sitting in a bathroom sink', 'a cat drinking
water from a kitchen sink', 'a cat playing in a sink with a
dripping faucet', 'a cat lying in a sink in a tiled bathroom', 'a
cat hiding in a sink']

6. H6_sitting positions: ['a cat sitting upright on a couch', 'a
cat sitting on a window sill', 'a cat sitting in a box', 'a cat
sitting on a pillow', 'a cat sitting in a sunlit spot']
```

Figure 21: LADDER discovers slices for biased attributes in RN Sup IN1k-based classifier for *cat* classification in **MetaShift** dataset. This figure details the slice discovery process for biased attributes involving sentence analysis, hypothesis generation by an LLM, and the model's performance on slices where attributes are present or absent, demonstrating how biases affect classifier accuracy. We highlight the hypothesis generated by LADDER that corresponds to the ground truth biased attribute (*e.g.,* `indoor` for cat) in **yellow**.

① Sentences indicating the biased attributes

1. a man walking a group of dogs on a **leash**
2. a group of people walking on the **beach** with their dogs
3. a dog playing in the water at the **beach**
4. a black and white dog playing with a toy in the **grass**
5. a dog is walking on the beach near the **water**
6. a group of men standing around a dog on a **leash**
7. a person walking a dog on a **leash**
8. a group of dogs playing in the **grass**
9. a dog walking on the **beach**
10. a woman **riding** a **surf board** with a dog
11. a man walking down the **street** holding a dog
12. a woman walking a dog on a **leash**
....

③ Performance of the model on slices where the biased attribute is present/absent

**H1: Objects related to sports:**

Present: 98.0 %
Absent: 84.9 %

**H2: Beach environments:**

Present: 99.4 %
Absent: 82.1 %

**H3: Dogs in motion:**

Present: 99.4 %
Absent: 82.4 %

**H4: Dogs with objects in their mouths:**

Present: 99.2 %
Absent: 82.2 %

**H5: Dogs on leashes :**

Present: 99.2 %
Absent: 81.7 %

② Hypotheses with biased attributes generated by LLM

```
Hypotheses:
The classifier is making mistake as it is biased toward:
H1: objects related to sports, H2: beach environments, H3:
in motion, H4: dogs with objects in their mouths, H5: dogs
leashes

Prompt to test each hypotheses:
1. H1_objects related to sports: ['A dog sitting next to a tennis
racket on a tennis court.', 'A dog running with a baseball bat in
its mouth on a field.', 'A dog standing next to a basketball on a
court.', 'A dog lying next to a soccer ball on a grass field.',
'A dog jumping to catch a frisbee in a park.']

2. H2_beach environments: ['A dog running on the beach near the
ocean.', 'A dog playing in the sand at the beach.', 'A dog
standing on a surfboard in the water.', 'A dog and a person
walking along the shore of a beach.', 'A dog digging a hole in
the sand at the beach.']

3. H3_dogs in motion: ['A dog running rapidly in a field chasing
a ball.', 'A dog jumping over a hurdle in an agility course.', 'A
dog chasing its tail in a park.', 'A dog sprinting along a street
next to a moving car.', 'A dog leaping into a pond to fetch a
stick.']

4. H4_dogs with objects in their mouths: ['A dog carrying a
newspaper in its mouth coming towards the porch.', 'A dog holding
a frisbee in its mouth ready to play.', 'A dog fetching a stick
in its mouth from the water.', 'A dog with a slipper in its mouth
greeting its owner.', 'A dog carrying a ball in its mouth during
a walk in the park.']

5. H5_dogs on leashes: ['A person walking a dog on a leash down a
busy street.', 'A dog on a leash sitting patiently at a bus
stop.', 'A dog on a leash interacting with another dog in a
park.', 'A dog on a leash being trained by its owner in obedience
class.', 'A dog on a leash waiting outside a store while its
owner shops.']
```

Figure 22: LADDER discovers slices for biased attributes in RN Sup IN1k-based classifier for *dog* classification in **MetaShift** dataset. This figure details the slice discovery process for biased attributes involving sentence analysis, hypothesis generation by an LLM, and the model's performance on slices where attributes are present or absent, demonstrating how biases affect classifier accuracy. We highlight the hypothesis generated by LADDER that corresponds to the ground truth biased attribute (*e.g.,* `outdoor` for cat) in **yellow**.

**Performance of the model on slices where the biased attribute is present/absent**

H1: Scattered calcifications:

Present: 83.5 %
Absent: 72.8 %

H2: Bilateral occurrences:

Present: 82.1 %
Absent: 77.9 %

H3: Multiple densities:

Present: 83.2 %
Absent: 73.5 %

H4: Benign appearances:

Present: 80.4 %
Absent: 77.3 %

H5: Vascular calcifications:

Present: 88.6%
Absent: 65.2 %

**Hypotheses with biased attributes generated by LLM**

Hypotheses:
The classifier is making mistake as it is biased toward:
H1: scattered calcifications
H2: bilateral occurrences
H3: the presence of multiple densities
H4: the description of benign appearances
H5: the mention of vascular calcifications

Prompt to test each hypotheses:
1. H1_scattered calcifications:
 ['Images showing scattered calcifications indicate positive cancer cases.',
'Scattered calcifications are a sign of breast cancer in mammograms.', 'Positive
cancer cases often present scattered calcifications in imaging.', 'Scattered
calcifications in a mammogram are typically associated with cancer.', 'Detecting
scattered calcifications is crucial for diagnosing cancer in patients.']

2. H2_bilateral occurrences:
['Bilateral occurrences in mammogram images indicate a positive cancer
diagnosis.', 'Cancer cases often show bilateral changes in breast tissue.',
'Bilateral symptoms in mammograms are significant for identifying cancer.', 'The
presence of bilateral abnormalities suggests cancer in mammogram analysis.',
'Images with bilateral occurrences are frequently linked to positive cancer
findings.']

3. H3_multiple densities:
 ['Multiple densities in mammograms are indicative of positive cancer cases.',
'The presence of multiple densities in breast imaging suggests cancer.', 'Cancer
in mammograms is often associated with multiple densities.', 'Identifying multiple
densities in mammograms is key to cancer detection.', 'Multiple densities in
imaging are a critical marker of breast cancer.']

4. H4_benign appearances:
['Descriptions of benign appearances in mammograms can mislead cancer detection.',
'Benign-looking features in mammograms may still represent cancer cases.', 'Cancer
detection is challenged by benign appearances in imaging.', 'Mammograms with
benign appearances might still be positive for cancer.', 'Identifying cancer in
the presence of benign appearances in mammograms is crucial.']

5. H5_vascular calcifications:
['Vascular calcifications in mammograms are often overlooked in cancer
diagnosis.', 'The presence of vascular calcifications can indicate underlying
cancer.', 'Cancer cases may present with vascular calcifications in mammograms.',
'Detecting cancer in the presence of vascular calcifications is essential.',
'Vascular calcifications in mammograms should be carefully evaluated for cancer.']

Figure 23: LADDER discovers slices for biased attributes for *cancer* classification in **RSNA-Mammo** dataset. This figure details the slice discovery process for biased attributes involving sentence analysis, hypothesis generation by an LLM, and the model's performance on slices where attributes are present or absent, demonstrating how biases affect classifier accuracy. We highlight the hypothesis generated by LADDER that corresponds to the ground truth biased attribute (*e.g.,* calcification for cancer) in **yellow**.

❶ Sentences indicating the biased attributes

1. a duck swimming in the **ocean** with a blue sky and **clouds** in the background
2. a seagul sitting on a rock in the **ocean**
3. a seagul sitting on a **dock** with **boats** in the background
4. a seagul catching a fish in the **ocean**
5. a seagul sits on the **water** in front of a container ship at night
6. a seagul sitting on a **rock** in front of a lighthouse at **sunset**
7. a seagul flying over the **ocean** with **rocks** and cliffs in the background
8. a seagul flying over the **ocean** at **sunset**
9. a seagul in the **water** with its wings spread out
10. a seagul flying over the **beach** with a **blue** sky in the background
11. a duck on the **beach** with a dark sky in the background
12. a seagul flying over the **water** with a fish in it's mouth
.....

❸ Performance of the model on slices where the biased attribute is present/absent

**H1: Specific bird activities like swimming or flying:**

**Present: 89.2 %**
**Absent: 58.6 %**

**H2: Specific times of the day like sunset or night:**

**Present: 83.7 %**
**Absent: 63.1 %**

**H3: Specific backgrounds like ocean or beach:**

**Present: 57.7 %**
**Absent: 90.3 %**

**H4: Presence of other objects like boats or rocks:**

**Present: 57.0 %**
**Absent: 91.1 %**

**H5: Weather conditions like cloudy or sunny:**

**Present: 83.2 %**
**Absent: 65.5 %**

❷ Hypotheses with biased attributes generated by LLM

```
Hypotheses:
The classifier is making mistake as it is biased toward:
H1: specific bird activities like swimming or flying
H2: specific times of the day like sunset or night
H3: specific backgrounds like ocean or beach
H4: presence of other objects like boats or rocks
H5: weather conditions like cloudy or sunny

Prompt to test each hypotheses:
1. H1_specific bird activities like swimming or flying:
['a bird swimming in the ocean', 'a bird flying over the beach',
'a bird catching a fish while flying', 'a bird diving into
water', 'a bird flapping its wings in the water']

2. H2_specific times of the day like sunset or night:
['a bird on the beach at sunset', 'a bird flying against a
sunset background', 'a bird sitting on a rock at night', 'a bird
under the colorful sky of sunset', 'a bird on the beach under
moonlight']

3. H3_specific backgrounds like ocean or beach:
['a bird with the ocean in the background', 'a bird standing on
the sandy beach', 'a bird flying over the waves', 'a bird perched
on a beach rock', 'a bird with the sea and beach in the
background']

4. H4_presence of other objects like boats or rocks:
['a bird on a boat in the ocean', 'a bird next to a rock in the
water', 'a bird flying over a boat', 'a bird standing next to a
boat on the beach', 'a bird perched on a rock by the sea']

5. H5_weather conditions like cloudy or sunny:
['a bird flying in a cloudy sky', 'a bird on the beach on a sunny
day', 'a bird in flight on a clear sunny day', 'a bird perched
under a cloudy sky', 'a bird on the water during a sunny
afternoon']
```

Figure 24: LADDER discovers slices for biased attributes in ViT Sup IN1k-based classifier for *waterbird* classification in **Waterbirds** dataset. This figure details the slice discovery process for biased attributes involving sentence analysis, hypothesis generation by an LLM, and the model's performance on slices where attributes are present or absent, demonstrating how biases affect classifier accuracy. We highlight the hypothesis generated by LADDER that corresponds to the ground truth biased attribute (*e.g.,* water for waterbirds) in **yellow**.

**❶ Sentences indicating the biased attributes**

```
1. a yellow bird perched on a branch in a bamboo forest
2. a bird perched on a tree branch in a bamboo forest
3. a cardinal bird in a bamboo forest
4. a bird perched on a tree in a bamboo forest
5. a bird perched on a branch in a bamboo forest
6. a bird sitting on a branch in a bamboo forest
7. a bird perched on top of a tree in a bamboo forest
8. a bird sitting on top of a tree in a bamboo forest
9. a bird perched on a bamboo tree in a bamboo forest
10. a yellow bird in the bamboo forest
11. a bird standing on a path in a bamboo forest
12. a bird perched on top of a bamboo tree in a bamboo forest
                        …
```

**❸ Performance of the model on slices where the biased attribute is present/absent**

H1: **Bamboo forest:**

**Present: 98.4 %**
**Absent: 75.2 %**

H2: bird perched on a branch:

**Present: 97.2 %**
**Absent: 71.6 %**

H3: Bird sitting on top of a tree:

**Present: 95.6 %**
**Absent: 77.6 %**

H4: Bird in the middle of a forest:

**Present: 95.6 %**
**Absent: 77.5 %**

H5: Yellow bird:

**Present: 95.2 %**
**Absent: 77.7 %**

**❷ Hypotheses with biased attributes generated by LLM**

**Hypotheses:**
The classifier is making mistake as it is biased toward:
H1: bamboo forest
H2: bird perched on a branch
H3: bird sitting on top of a tree
H4: bird in the middle of a forest
H5: yellow bird

**Prompt to test each hypotheses:**
**1. H1_bamboo forest:**
['A bird flying through a dense bamboo forest', 'A small bird perched in a lush bamboo forest', 'A colorful bird hidden among bamboo trees', "A bird's silhouette against the bamboo forest at sunset", 'A bird chirping from within a thick bamboo grove']

**2. H2_bird perched on a branch:**
['A bird perched quietly on a tree branch', 'A small bird holding onto a swaying branch', 'A bird perched on a branch overlooking the forest', 'A bird perched on a leafless branch in winter', 'A bird resting on a branch during a rainy day']

**3. H3_bird sitting on top of a tree:**
['A bird sitting on the highest branch of a tall tree', 'A bird surveying its surroundings from atop a tree', 'A bird nesting on the top branches of a tree', 'A bird perched majestically on top of a tree at dawn', 'A bird catching the morning sun on the top of a tree']

**4. H4_bird in the middle of a forest:**
['A bird camouflaged among the forest leaves', 'A bird singing in the heart of the forest', 'A bird foraging on the forest floor', 'A bird flitting through the dense forest undergrowth', 'A bird perched quietly in the deep forest']

**5. H5_yellow bird:**
['A bright yellow bird standing out against green leaves', 'A yellow bird fluttering from flower to flower', 'A small yellow bird perched on a garden fence', 'A yellow bird in flight against a clear blue sky', 'A yellow bird preening its feathers']

Figure 25: LADDER discovers slices for biased attributes in ViT Sup IN1k-based classifier for *landbird* classification in **Waterbirds** dataset. This figure details the slice discovery process for biased attributes involving sentence analysis, hypothesis generation by an LLM, and the model's performance on slices where attributes are present or absent, demonstrating how biases affect classifier accuracy. We highlight the hypothesis generated by LADDER that corresponds to the ground truth biased attribute (*e.g.,* land for waterbirds) in **yellow**.

**❶ Sentences indicating the biased attributes**

1. a woman with **long blonde** hair smiling and **wearing a black top**
2. a woman with **long blonde** hair standing in front of **a red carpet**
3. a woman with **long blonde** hair standing on **a red carpet**
4. a woman with **long blonde** hair is smiling and holding a pillow
5. a girl with **long blonde** hair in a **black** frame on a **white** background
6. a woman with blonde hair standing on a **red carpet**
7. a woman with **long blonde** hair standing in front of a stone wall
8. a woman with **long blonde** hair posing in front of a **christmas tree**
9. a woman with **long blonde** hair wearing a **black** dress
10. a woman with **long blonde** hair **smiling** at the camera
11. a young woman with **long blonde** hair smiles at the camera
12. a woman with long **blonde hair** and a **black jacket** smiles at the camera
....

**❸ Performance of the model on slices where the biased attribute is present/absent**

H1: black dress:
Present: 92.4 %
Absent: 70.9 %

H2: Woman wearing a red dress:
Present: 92.0 %
Absent: 71.3 %

H3: Woman wearing a blue dress:
Present: 92.0 %
Absent: 71.3 %

H4: white dress:
Present: 92.7 %
Absent: 70.6 %

H5: black top:
Present: 99.2 %
Absent: 70.9 %

**❷ Hypotheses with biased attributes generated by LLM**

Hypotheses:
The classifier is making mistake as it is biased toward:
H1: black dress
H2: woman wearing a red dress
H3: a blue dress
H4: white dress
H5: black top
H6: blue top
H7: woman with red top
H8: white top
H9: black jacket
H10:green dress

Prompt to test each hypotheses:
1. **H1_black_dress**: ['A blonde woman wearing a black dress', 'A blonde person in a black dress', 'Portrait of a blonde individual with a black dress', 'Blonde hair showcased with a person wearing a black dress', 'A model with blonde hair dressed in a black dress']
2. **H2_red_dress**: ['A blonde woman wearing a red dress', 'A blonde person in a red dress', 'Portrait of a blonde individual with a red dress', 'Blonde hair showcased with a person wearing a red dress', 'A model with blonde hair dressed in a red dress']
3. **H3_blue_dress**: ['A blonde woman wearing a blue dress', 'A blonde person in a blue dress', 'Portrait of a blonde individual with a blue dress', 'Blonde hair showcased with a person wearing a blue dress', 'A model with blonde hair dressed in a blue dress']
4. **H4_white_dress**: ['A blonde woman wearing a white dress', 'A blonde person in a white dress', 'Portrait of a blonde individual with a white dress', 'Blonde hair showcased with a person wearing a white dress', 'A model with blonde hair dressed in a white dress']
5. **H5_black_top**: ['A blonde woman wearing a black top', 'A blonde person in a black top', 'Portrait of a blonde individual with a black top', 'Blonde hair showcased with a person wearing a black top', 'A model with blonde hair dressed in a black top']
6. **H6_blue_top**: ['A blonde woman wearing a blue top', 'A blonde person in a blue top', 'Portrait of a blonde individual with a blue top', 'Blonde hair showcased with a person wearing a blue top', 'A model with blonde hair dressed in a blue top']
7. **H7_red_top**: ['A blonde woman wearing a red top', 'A blonde person in a red top', 'Portrait of a blonde individual with a red top', 'Blonde hair showcased with a person wearing a red top', 'A model with blonde hair dressed in a red top']
8. **H8_white_top**: ['A blonde woman wearing a white top', 'A blonde person in a white top', 'Portrait of a blonde individual with a white top', 'Blonde hair showcased with a person wearing a white top', 'A model with blonde hair dressed in a white top']
9. **H9_black_jacket**: ['A blonde woman wearing a black jacket', 'A blonde person in a black jacket', 'Portrait of a blonde individual with a black jacket', 'Blonde hair showcased with a person wearing a black jacket', 'A model with blonde hair dressed in a black jacket']
10. **H10_green_dress**: ['A blonde woman wearing a green dress', 'A blonde person in a green dress', 'Portrait of a blonde individual with a green dress', 'Blonde hair showcased with a person wearing a green dress', 'A model with blonde hair dressed in a green dress']

Figure 26: LADDER discovers slices for biased attributes in ViT Sup IN1k-based classifier for *blond* classification in **CelebA** dataset. This figure details the slice discovery process for biased attributes involving sentence analysis, hypothesis generation by an LLM, and the model's performance on slices where attributes are present or absent, demonstrating how biases affect classifier accuracy. We highlight the hypothesis generated by LADDER that corresponds to the ground truth biased attribute (*e.g.,* woman for blond) in **yellow**.

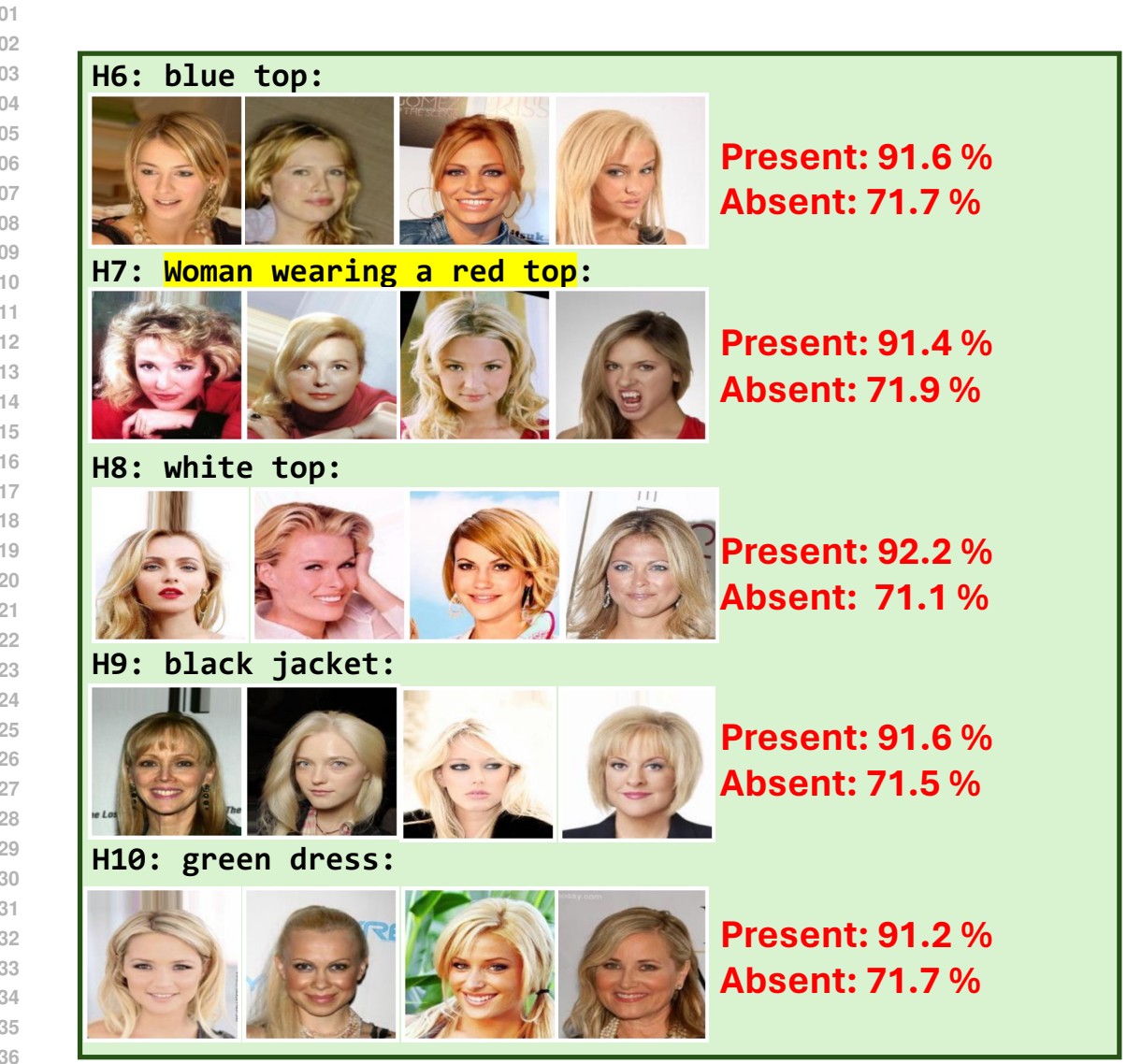

Figure 27: Peformance of the ViT Sup IN1k-based classifier on additional slices where attributes are present/absent for *blond* classification in **CelebA** dataset.

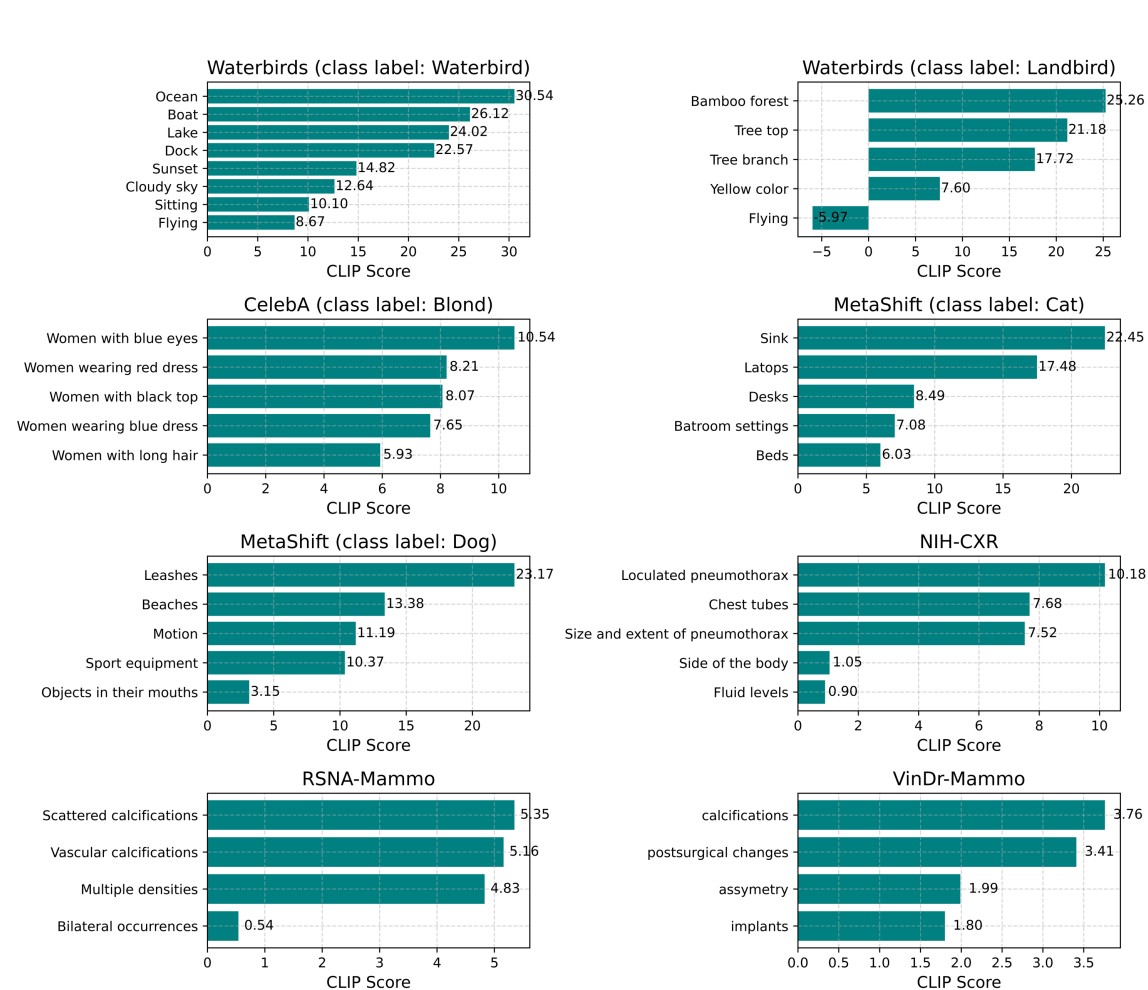

Figure 28: CLIP Score(Appendix A.5) for various attributes extracted from the hypotheses by LADDER. CLIP scores of the attributes are high signifying that they induce biases on the correctly classified samples.

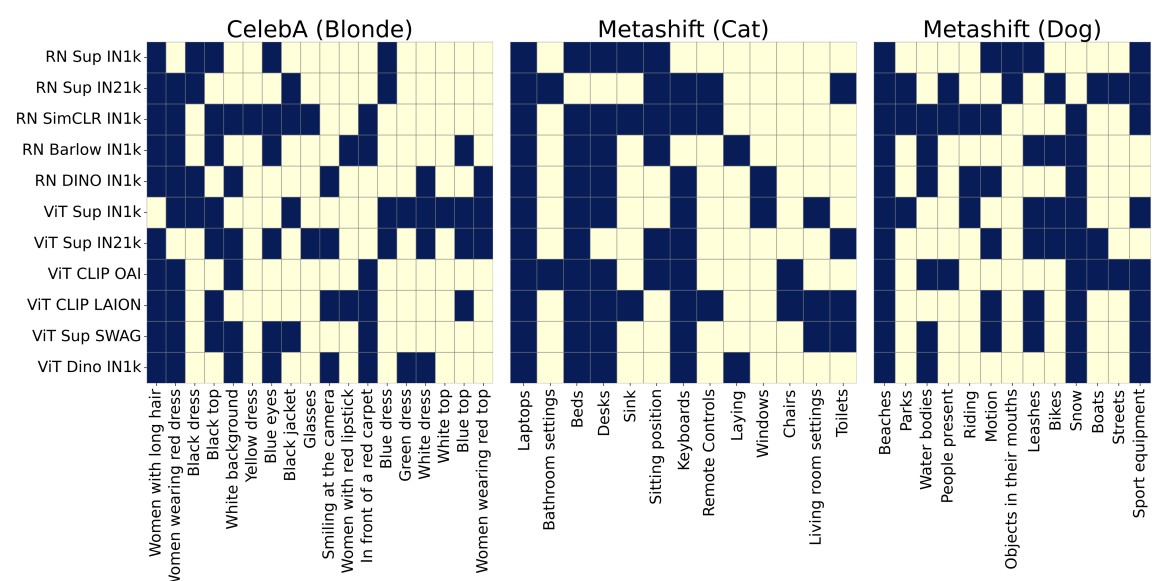

Figure 29: LADDER accurately captures various sources of bias, regardless of the underlying architectures or pre-training methods for the CelebA and Metashift datasets. Bright colors indicate attributes in LADDER's hypotheses, while light colors indicate their absence.

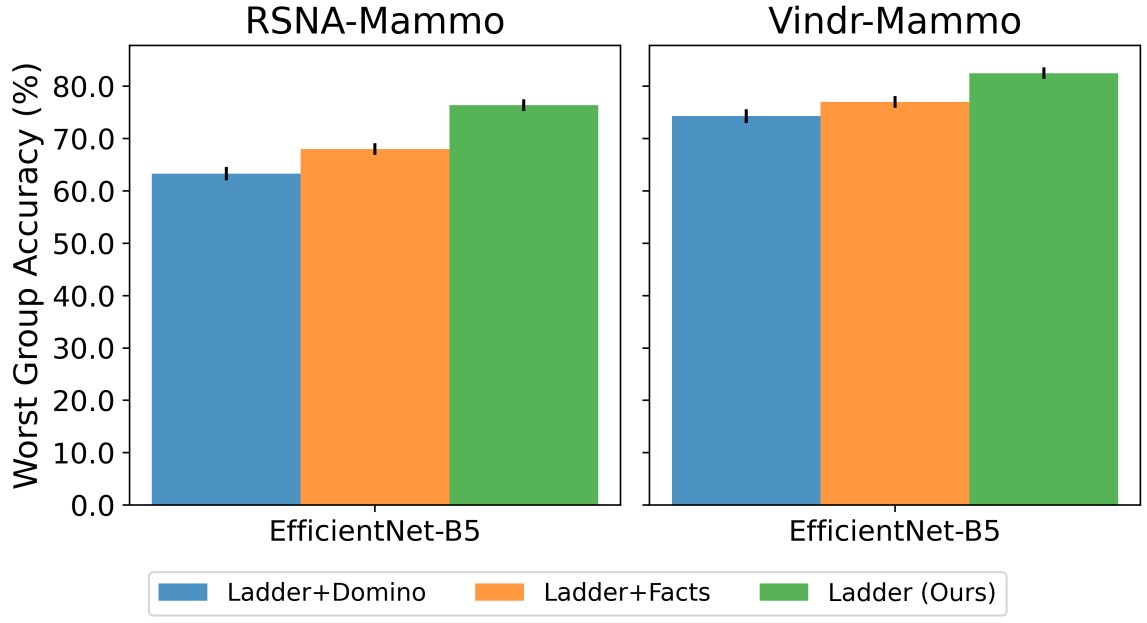

Figure 30: LADDER improves WGA compared to other bias mitigation methods for RSNA-Mammo and VinDr-Mammo datasets.

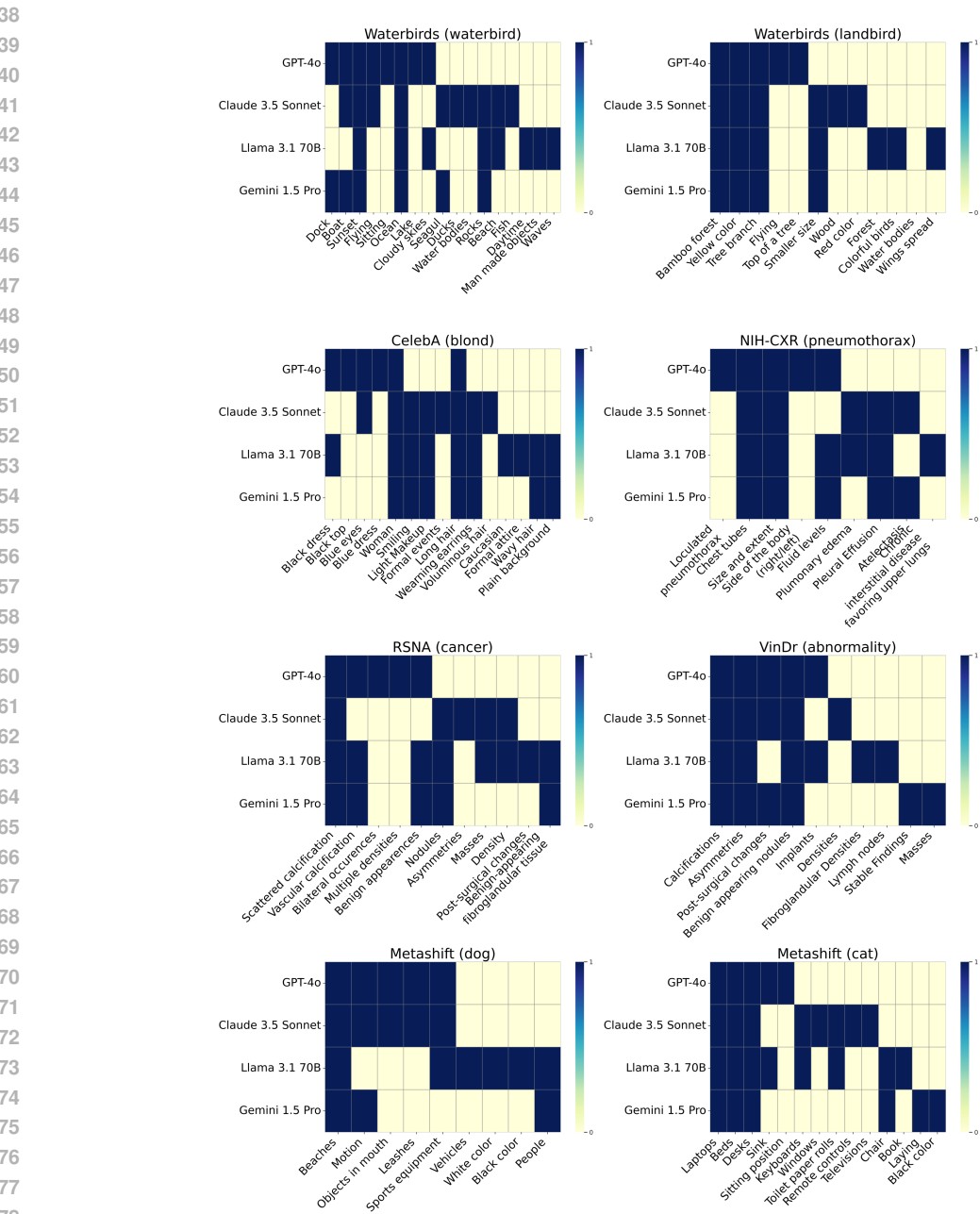

Figure 31: Ablation 2: Attributes identified by different LLMs while generating hypotheses across datasets for bias identification: RN Sup IN1k for natural images and CXRs, and EN-B5 for mammograms. Each LLM (GPT-4o, Claude 3.5 Sonnet, LlaMA 3.1 70B, Gemini 1.5 Pro) focuses on distinct attributes, yet the overall hypotheses are consistent across datasets, showing LADDER's robust bias detection. Bright colors indicate attributes in LADDER's hypotheses, while light colors indicate their absence.

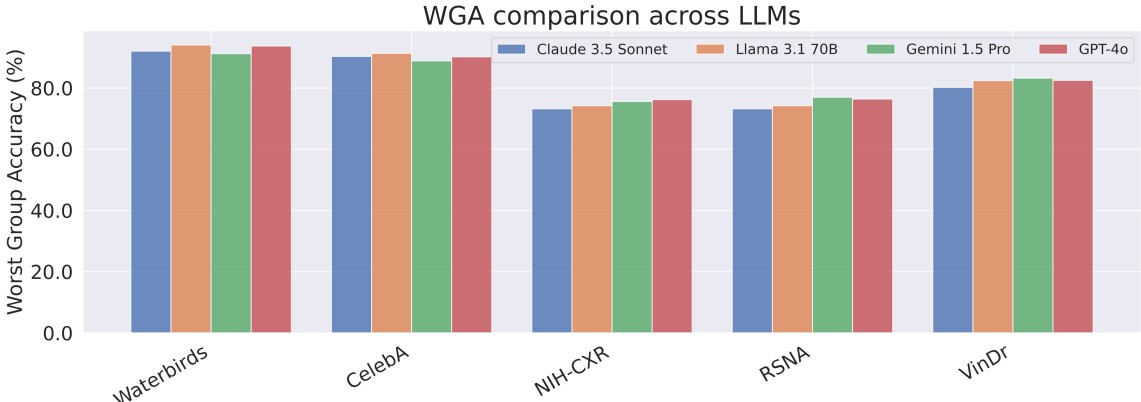

Figure 32: Ablation 3: Worst Group Accuracy (WGA) comparison across different LLMs for bias mitigation by LADDER in multiple datasets with RN Sup IN1k as the classifier for natural images and CXRs, and EN-B5 for mammograms. LADDER effectively reduces biases across all LLMs, maintaining consistent WGA performance.

