# OpenReview forum: "Ladder: Language Driven Slice Discovery and Error Rectification"
_ICLR.cc/2025/Conference — ICLR 2025 Conference Withdrawn Submission_

### Official Review · Reviewer_fv6Q · 2024-11-03

**Soundness:** 3
**Presentation:** 3
**Contribution:** 2
**Rating:** 5
**Confidence:** 4

**Summary:**

The paper introduces LADDER, a novel method for diagnosing and mitigating biases in image classification models. Traditional methods for error slice discovery, such as clustering or using discrete attributes, face limitations like incoherent slices, incomplete coverage of error patterns, and lack of complex reasoning, especially in specialized domains like radiology. LADDER addresses these issues by leveraging the flexibility of natural language and harnessing the latent domain knowledge and reasoning capabilities of large language models (LLMs).

Specifically, for a given class label, LADDER uses image captions (or generates them using vision-language models) and encodes both images and text into a joint embedding space using models like CLIP. It then computes the difference in mean representations between correctly classified and misclassified samples in the image embedding space. By retrieving top sentences from the text embedding space that align with these mean representations, LADDER captures the primary misalignments between correct and incorrect classifications. These sentences are fed into an LLM to identify biased attributes, forming coherent error slices without the need for clustering.

To mitigate the identified biases, LADDER generates pseudo-attributes from the discovered hypotheses and reweights training examples accordingly. This approach does not require explicit attribute annotations or prior knowledge of biases, allowing for error mitigation across all biases rather than focusing on the worst-performing group. The authors rigorously evaluate LADDER on six datasets, including natural and medical images, comparing it against over 200 classifiers with diverse architectures and pretraining strategies. The results demonstrate that LADDER consistently outperforms existing baselines in both bias discovery and mitigation.

**Strengths:**

1. The paper presents a creative approach that combines natural language flexibility with the reasoning power of LLMs to address bias in image classification, moving beyond the limitations of traditional clustering or attribute-based methods. By leveraging LLMs' latent domain knowledge, LADDER effectively identifies and explains complex biases, making it particularly valuable in specialized fields like radiology where domain expertise is crucial.
2. The authors conduct extensive experiments across six diverse datasets, testing over 200 classifiers with various architectures, pretraining strategies, and LLMs. This thorough evaluation strengthens the validity and applicability of the proposed method. The paper is well-written and easy to follow, providing detailed comparisons and ablation studies that offer deep insights into the method's performance and underlying mechanisms.

**Weaknesses:**

1. The method heavily relies on the availability of image captions or the effectiveness of vision-language models to generate them. It also depends on joint image-language embeddings like CLIP, which may introduce additional complexity and potential limitations if these models are biased or not well-suited to the specific data. For example, the reliance on LLMs to identify biased attributes from top sentences introduces uncertainties. If the spurious features are subtle, adversarial, or not easily recognizable from captions, the LLM may struggle to identify them.
2. If the pretrained models (e.g., CLIP, LLMs) used in LADDER are themselves biased, there is a risk that these biases could propagate through the analysis, affecting the identification and mitigation of biases in the target classifier.
3. The approach assumes that mean representations in the embedding space sufficiently capture the central tendencies of correct and incorrect classifications. However, image distributions within a single class may be multimodal due to varying underlying attributes. It's unclear how well the method handles such complexities or overlapping distributions.

**Questions:**

1. How many samples are required for the LLM to reliably recognize spurious or biased attributes? From the examples shown in Figure 1, I think GPT-4o should to be able to identify these attributes with very few correct and incorrect examples. Could the authors provide insights or analyses on the minimum sample size needed for robust bias detection, especially in cases where biases are less apparent?
2. In situations where image captions are unavailable or when captions do not cover all spurious features (e.g., images with subtle or adversarial features not readily describable), how effective is LADDER? Can the method be adapted to function without relying on captions, or is there a way to enhance its robustness in such contexts?
3. The Waterbirds dataset is synthetically generated -- it should not be referred to as "natural images".
4. Missing reference to which identifies spurious bias clusters without assuming access to external VLM / LLM models ( https://arxiv.org/pdf/2204.13749)

**Details Of Ethics Concerns:**

This paper presents an automated method for bias identification and mitigation in image classification models using LADDER. While the approach demonstrates potential for enhancing model fairness and performance, there are ethical considerations that should be addressed:

1. Risk of Misidentifying Biases:
    * False Positives: The automated bias identification process may incorrectly label legitimate, causal features as biases. This misidentification could lead to the undesired suppression of important features that are critical for accurate predictions.
    * Labeling Errors: Without human oversight, labeling mistakes or anomalies in the data may be misconstrued as biases, potentially leading to improper model adjustments.
2. Absence of Human-in-the-Loop:
    * Ethical Oversight: The completely automated process lacks a mechanism for human experts to review and validate the identified biases. Human judgment is crucial to ensure that the biases being corrected are genuine and that mitigation strategies are appropriate.
    * Accountability: Without human intervention, it becomes challenging to hold any entity accountable for decisions made by the model, especially in sensitive applications like radiology.
3. Reliance on Potentially Biased Models:
    * The method depends on pretrained models like CLIP and LLMs, which may inherit existing biases from their training data. These biases could influence both the identification and mitigation processes, undermining the fairness goals.
4. Ethical Responsibility in Specialized Domains:
    * In fields like radiology, incorrect bias mitigation could have serious implications for patient care. Ethical considerations are paramount, and decisions must be carefully evaluated by domain experts.

---

### Official Review · Reviewer_xWqC · 2024-11-04

**Soundness:** 2
**Presentation:** 3
**Contribution:** 2
**Rating:** 5
**Confidence:** 3

**Summary:**

The authors tackle the problem of error slice discovery using natural language by proposing the LADDER framework. Given a corpus of text descriptions, LADDER first uses the latent space of a VLM to find top-k text embeddings which have the highest similarity with the difference in mean representations between correct and misclassified samples. These candidate captions are then passed to an LLM which extracts hypotheses. The authors also propose a mitigation strategy where images which do not contain these hypotheses are upweighted. The authors evaluate their method on spurious correlation and medical imaging datasets, finding that they discover better slices than the baselines, and that their mitigation strategy outperform common subpopulation shift methods.

**Strengths:**

- The problem is important and well-motivated.
- The authors evaluate a wide array of base architectures, and a large set of mitigation baselines.
- The authors evaluate their method on multiple real-world medical datasets.

**Weaknesses:**

1. The method heavily relies on a corpus of text captions which sufficiently describes all aspects of the images which could potentially correspond to error slices. The authors use a captioning model for this in their experiments, but do not conduct any ablations. The authors should also present results in MIMIC-CXR using generated radiology reports, where the quality of the text corpus may be significantly worse than for natural (or potentially mammography) images.

2. The proposed method uses the convention that images which do not contain a certain attribute have higher error. How would the method work if there exists an error slice that _contains_ a certain attribute?

3. In Table 4, in several instances, LADDER exhibits WGA higher than mean accuracy. I do not see how this is possible.

4. All of the components of the pipeline (BLIP-captioner and VLMs) are highly specific to the image domain, which limits the utility of the method. It seems like the method should be easily adaptable to detecting errors in text classification as well.

5. All of the datasets which the authors evaluate on are binary classification datasets. I would like to see a larger-scale multi-class classification dataset such as ImageNet.

6. I don't find the higher performance of LADDER over DOMINO and FACTS particularly compelling, as neither of the two baselines require access to an LLM, which LADDER does. Does LADDER still outperform DOMINO and FACTS in Figure 3 if Llama-3.1 is used instead of GPT-4o?

7. The method does not seem to adjust for sample size of the error slice. Is there a failure mode where the method outputs highly specific slices that contain very few examples? The authors should also show sample size in their results (e.g. Figure 5).

**Questions:**

1. When discovering slices for downstream error mitigation, is the error slicing done on the training set? If so, how would the framework work for overparameterized models that have (close to) zero training error? If it is done on a held-out set, this seems unfair to the baseline mitigation methods as they do not have access to this set.

---

### Official Review · Reviewer_btVw · 2024-11-07

**Soundness:** 2
**Presentation:** 2
**Contribution:** 2
**Rating:** 3
**Confidence:** 3

**Summary:**

The paper makes two key contributions:
* It presents Ladder, a method for characterizing "error slices" (subsets of the data on which model performance is >10% worse than the aggregate) using natural language. (E.g., it produces hypotheses like "images with boats characterize error slices").
* It shows how to use these hypotheses as logits for reweighing members of those error classes during training to mitigate the error.

**Strengths:**

* The problem is very neat -- using natural language to interpret where models under-perform and why they under-perform on those regions.
* The means of using pseudo-labelling to correct for the error slices is clever and intuitive. The authors are to be sincerely commended for this extension to their work.
* The results are interesting -- while I do have questions (and some concerns) regarding the methodological details, it is promising to see that the method picks up on salient and plausible features that may define error slices.

**Weaknesses:**

1. The paper is sloppily written. Below is a non-exhaustive list of examples, in consecutive order, to illustrate the point. This paper would benefit from a significant writing overhaul to meet the quality bar for this conference.
* _There are two errors here, identified with asterisks: (1) It should read *LLMs'*, as it refers to the latent domain knowledge of the class of language models. (2) It should read *identify* to preserve parallel form._
> employing LLM's* latent _domain knowledge_ and advanced reasoning to analyze sentences and derive testable hypotheses directly, identifying* biased attributes, and form coherent error slices. [Lines 018 - 020]
* _This is the opening sentence; however, at this point, the reader does not know what an "error slice" is. This makes the organization confusing. Perhaps the paper could begin by explaining what an error slice is, then explaining why it is important._
> Discovering error slides in models is essential for mitigating their limitations. [Line 031]
* There is inconsistency in how enumerated lists are numbered _within the first paragraph_. The first list of "key issues" uses 1), 2), 3), for numbering; the second list of how Ladder addresses the issues uses (1), (2), etc.
> Bolding is interchangeably used for sub-headings and for emphasis (e.g., Line 180 -- 183; lines 210 -- 211). Rather than condensing headings into paragraphs (which is hard to parse), condense / clean up the writing and use normal LaTeX headers.

> Vague language ("Ladder finds error slices where $f$ underperforms and **fixes it**", Line 095). "fixes it" is vague and causal writing; additionally, this is in the _notation_ section of the paper -- making this sentence very out of place.

(These are all in the opening several paragraphs -- the remainder of the document contains many similar such issues).

More broadly on writing, there are three higher-level concerns I have:
* The experimental details are hard to follow. Too much emphasis is placed on specific model architectures (see: most of page 5), and too little on cleanly detailing the overall method as in Figure 2 (see, for example, my second question below -- that this type of information is not apparent from the manuscript is a major weakness).
* The organization is lacking. The Introduction seems to describe related work (around Lines 048), and does not clearly enumerate contributions ("Contributions" gives an overview of how the method works, rather than telling me how this paper builds atop prior literature in a way that hasn't been studied before). The notation section contains discussions of the results ("Ladder finds error slices where $f$ underperforms and **fixes it**"), as well as a summary of the method (the reference to Figure 2 and associated discussion on Line 98).
* A lot of pointers to appendices break flow -- e.g., to even remotely understand the experiment underlying Figure 1 (p. 2), the reader needs to reference the associated appendix. The manuscript could do more to communicate the core ideas clearly in the main body so that referencing the appendices is _useful, but not **required**_ to understand the work that was done.

2. The method doesn't appear to work very well in practical settings. Specifically, the hypotheses generated seem too vague to be useful. Consider Figure 5. Among others, Ladder generates the following natural language hypotheses to identify error slices. (I have enclosed my own commentary in italics after each hypothesis.
> **H1: Specific background elements like docks and boats (Present: 97.0%; Absent, 68.8%).** _"Docks and boats" are examples of "specific background elements". Therefore, H1 refers to any images that contain "specific background elements". It's not entirely clear what this means: I would imagine that most images where the bird is atop a background more complex than a plain monocolor background would include "specific background elements," even if those elements are sky, ocean, beach, land, etc. Moreover, I'm concerned that this specific hypothesis is an **artifact of the prompt / model,** as many of the "sentences indicating biased attributes" in Figure 5(c) describe both the foreground and the background: therefore, the hypothesis generation may well latch on to terms like "background" when defining error slices, even though this is a general characteristic of a description generated by the LLM. It is not specific to the error class in question._

> **H3: Specific actions like flying or sitting (Present: 97.3%, Absent, 68.6%).** _I struggle to envision a bird in the dataset that is pictured in a position that is neither "flying" nor "sitting". Perhaps swimming? Either way, this hardly seems a useful interpretation for defining an error slice._

> **H4: Presence of water bodies like oceans and lakes. (Present: 97.6%, Absent 68.2%)** _For a dataset called "waterbirds", I imagine that most of the birds are pictured without bodies of water present.

3. Precision@10 seems to be one valuable metric, certainly, but is **hardly the only one** that should be used to compare Ladder against slice discovery baselines. I would imagine there are significantly more than 10 images per slice -- in that context, perhaps accuracy / precision / recall / F1 / AUC are better metrics? (Do feel free to adjust the statistic based on the mismatch between the positive (in slice) and negative (not in the slice) classes; but the point is that Precision@10 as presented in Figure 3 seems to provide a very incomplete picture of the relative performance of each measure (especially when there are (a) no confidence intervals, and (b) there is low granularity since it's definitionally rounded to the nearest 0.1).

4. Without further experiments, I'm unconvinced that Ladder successfully generates the _testable hypotheses_ (claimed on Lines 019 and 075) that are claimed by the paper. Looking at Figure 7, two out of the five hypotheses are concerned with the induced spurious correlation, but the authors do not appear to suggest (a) any _experimental tests_ that could deduce whether H1 and H3 are correct (rather than, say, a hallucination), or (b) any tests to confirm the relative validity of H1 and H3 with respect to the other hypotheses.

**Questions:**

1. The biased attributes detected by Ladder seem to significantly vary across different architectures and datasets (Figure 4). If it were invariant, I would expect vertical columns of blue and yellow. The more scattered representations here suggest that the performance does significantly vary. However, the authors claim that Ladder's biased attribute detection is invariant across different architectures and datasets. Why is this?
2. In Figure 5, it's not clear how the ground truth of whether a biased attribute is present/absent is determined (e.g., when saying that the model achieves 97% accuracy on images with "specific background elements, like docks and boats" and 68.8% accuracy when those elements are not present. My main concern is that, if the ground truth is determined by the language model (e.g., whether the sentence associated with that image contains the keyword in question), this analysis is subject to bias wherein the ground truth depends on the prediction in question -- e.g. **it is possible that the performance gap between present/absent has less to do with what is actually in the picture, and more to do with what the LLM _detects_ is in the picture**. From what I can see this is not ruled out in the present analysis.
3. Is the method entirely restricted to vision models, or are there other kinds of models that it can work on (e.g., time series, tabular data, etc.)? It appears to be vision only, but the authors claim that it works with "any off-the-shelf supervised classifier" (Line 065).

---

### Official Review · Reviewer_dHoW · 2024-11-09

**Soundness:** 3
**Presentation:** 3
**Contribution:** 2
**Rating:** 5
**Confidence:** 2

**Summary:**

The paper presents a novel approach for error slice discovery and bias mitigation in machine learning models. The key contributions of the paper are as follows:
﻿
Language-Driven Slice Discovery: The paper introduces LADDER, a method that leverages natural language processing and Large Language Models (LLMs) to identify error slices in models. This approach addresses the limitations of current methods, which either produce incoherent slices or suffer from incomplete coverage due to missing attributes.
﻿
Integration of Domain Knowledge: LADDER incorporates the latent domain knowledge and reasoning capabilities of LLMs to analyze sentences and derive testable hypotheses directly. This allows for the identification of biased attributes and the formation of coherent error slices without the need for clustering, which is a departure from traditional clustering-based methods.
﻿
Bias Mitigation: Unlike existing methods that typically address only the worst-performing group, LADDER generates pseudo attributes from discovered hypotheses to mitigate errors across all biases. This is achieved without explicit attribute annotations or prior knowledge of biases.

In summary, the paper introduces a new framework that enhances the discovery and mitigation of model errors by utilizing the flexibility of natural language and the advanced reasoning capabilities of LLMs, offering a significant advancement over current slice discovery and bias mitigation techniques.

**Strengths:**

### Originality
- **Innovative Approach**: The paper introduces LADDER, a novel method that combines natural language processing with LLMs for error slice discovery and bias mitigation, representing a creative advancement in the field.
- **Application to New Domains**: The method is applied to both natural general and medical images, demonstrating its versatility and potential impact across different domains, including specialized fields like radiology.

### Quality
- **Rigorous Evaluations**: The paper provides extensive experimental evaluations on six diverse datasets, ensuring the method's effectiveness and robustness are thoroughly tested.
- **Comparison with Multiple Baselines**: LADDER is compared against over 200 classifiers with various architectures and pretraining strategies, which strengthens the credibility of the results.

### Clarity
- **Clear Structure**: The paper is well-organized, with a logical flow from introduction to methodology, experiments, and conclusions, making it easy to follow.
- **Comprehensive Explanation**: The methodology is explained clearly, with sufficient details on the technical aspects of LADDER, aiding in understanding its workings.

### Significance
- **Enhancing Model Interpretability**: The paper contributes to the interpretability of machine learning models by uncovering and mitigating biases, which is increasingly important for trust and adoption in high-stakes applications.

**Weaknesses:**

### Generalizability and Limitations
- **Dependence on Quality of Captions and VLR related Models**: The performance of LADDER is heavily reliant on the quality of available captions and the vision-language representation (VLR) related models (including CLIP). The paper could benefit from a discussion on how variations in these components might affect the outcomes. For example, analyzing the impact of different forms of VLR on the results, such as the alignment form of LLAVA instead of CLIP. This will also be of great help for future promotion

### Methodological Transparency
- **Opaque Use of LLMs**: The specific prompts and interactions with LLMs are not detailed extensively. Providing more transparency on how LLMs are queried and their responses are interpreted could strengthen the methodology section.

### Cost and Resource Intensity
- **Resource Requirements for LLMs**: The paper mentions the cost of using LLMs but does not discuss the trade-off between performance gains and computational costs, especially for smaller institutions or when scaling.

By addressing these weaknesses, the paper could provide a more comprehensive view of LADDER's capabilities, limitations, and potential impacts.

**Questions:**

1. How do variations in the quality of captions and the choice of vision-language representation models (e.g., CLIP vs. LLAVA) impact the performance and generalizability of LADDER? Can you provide empirical evidence or a comparative analysis to illustrate these effects?

2. Can you provide detailed information on the specific prompts and interactions used with large language models (LLMs) in your methodology? How do these prompts influence the LLMs' responses, and what measures are in place to ensure consistent and accurate interpretations of these responses?

3. Given the resource-intensive nature of using large language models (LLMs), what considerations and trade-offs did you encounter between the computational costs and performance gains of LADDER?

---

### Note · Authors · 2024-11-15

**Comment:**

We agree to withdraw. However we thank the reviewers for their feedback. We will shortly post a response to their concerns.

**Withdrawal Confirmation:**

I have read and agree with the venue's withdrawal policy on behalf of myself and my co-authors.